# Score-of-Mixture Training: One-Step Generative Model Training Made Simple via Score Estimation of Mixture Distributions

**Tejas Jayashankar**[* 1]  **J. Jon Ryu**[* 1]  **Gregory Wornell**[1]

## Abstract

We propose *Score-of-Mixture Training* (SMT), a novel framework for training one-step generative models by minimizing a class of divergences called the $\alpha$-skew Jensen–Shannon divergence. At its core, SMT estimates the score of mixture distributions between real and fake samples across multiple noise levels. Similar to consistency models, our approach supports both training from scratch (SMT) and distillation using a pretrained diffusion model, which we call *Score-of-Mixture Distillation* (SMD). It is simple to implement, requires minimal hyperparameter tuning, and ensures stable training. Experiments on CIFAR-10 and ImageNet 64×64 show that SMT/SMD are competitive with and can even outperform existing methods.

## 1. Introduction

Fast and efficient sampling is a key characteristic sought after in modern generative samplers. For many years, generative adversarial networks (GANs) (Goodfellow et al., 2014) set the benchmark for high-quality one-step generative sampling. However, due to the inherent training instabilities associated with discriminator training, attention has recently shifted toward diffusion-based generative models (Sohl-Dickstein et al., 2015; Ho et al., 2020; Karras et al., 2022b). These models trade-off sampling efficiency for more stable training and significantly improved downstream sample quality through iterative sampling.

More recently, the diffusion distillation approach has been studied as an appealing option to significantly reduce the number of sampling steps. Early work (Luhman & Luhman, 2021; Salimans & Ho, 2022; Meng et al., 2023; Berthelot et al., 2023) focused on training a student model with a lower sampling budget by condensing multiple teacher denoising steps into one. The most recent works on distillation improve performance further by leveraging a pretrained model for distribution matching via minimization of the reverse KL divergence (Luo et al., 2024a; Yin et al., 2024b;a; Salimans et al., 2024; Xie et al., 2024). While attractive, distillation approaches necessitate a pretrained diffusion model which adds a significant overhead on the required compute.

As yet another alternative, consistency models (Song et al., 2023; Song & Dhariwal, 2024b) and their variants (Kim et al., 2024) have been proposed for training few-step generative models from scratch by simulating the trajectories of the induced probability flow ODE (Song et al., 2020) of a diffusion process. While consistency models have demonstrated promising results in both distillation and training from scratch, training is sensitive to the choice of noise schedule and distance measure (Geng et al., 2025).

In this paper, we tackle the problem of training high-quality one-step generative models more directly, i.e., *without* simulating an iterative reverse diffusion process for sampling or leveraging a pretrained diffusion model during training. Starting from first principles of statistical divergence minimization, we show that a high-quality one-step generative model can be trained from scratch in a stable manner, via the multi-noise-level denoising score matching (DSM) technique (Vincent, 2011) used in diffusion models. We emphasize that we do not require a simulation of the reverse diffusion process in our framework.

The proposed framework achieves the best of several worlds: (1) a new, simple statistical divergence minimization framework without probability paths of ODE (like GAN), (2) stable training using denoising score matching (like diffusion models), (3) training from scratch without a pretrained diffusion model (like consistency models), and (4) near state-of-the-art one-step image generative performance (like GAN and consistency models). We also demonstrate that the proposed method can be extended to distill from a pretrained diffusion model, and can achieve performance similar to state-of-the-art methods for the same. See Table 1 for the overview of comparison.

---

[*]Equal contribution  [1]Department of Electrical Engineering and Computer Science, Massachusetts Institute of Technology (MIT), Cambridge, Massachusetts, USA. Correspondence to: Tejas Jayanshankar <tejasj@mit.edu>, J. Jon Ryu <jongha.ryu@gmail.com>.

*Proceedings of the 42$^{nd}$ International Conference on Machine Learning*, Vancouver, Canada. PMLR 267, 2025. Copyright 2025 by the author(s).

*Table 1.* Comparison of different generative modeling techniques capable of high-quality sample generation.

| Generative models | Training idea | Generation | Training stability | Require pretrained model? |
|---|---|---|---|---|
| GAN | minimizing JSD, with discriminator | one-step | unstable | N |
| Diffusion models | training multi-noise-level denoisers via DSM | multi-step | stable | N |
| Diffusion distillation | (mostly) minimizing reverse KLD (in DMD) | {one,few}-step | stable | Y |
| Consistency distillation | simulating trajectories of probability flow ODE | {one,few}-step | stable | Y |
| Consistency training | | | unstable | N |
| **Score-of-Mixture Training** (ours) | minimizing $\{\alpha\text{-JSD}\}_{\alpha \in [0,1]}$ with multi-noise-level | one-step | stable | N |
| **Score-of-Mixture Distillation** (ours) | training & scores of *mixtures* via DSM | | | Y |

The rest of the paper is organized as follows: In Sec. 2 we introduce the necessary background and related works central to our proposed method. In Sec. 3 we introduce our novel one-step generative modeling approach and in Sec. 4 we detail how our framework can be modified to perform diffusion distillation. We describe practical implementation details in both the latter sections and present experimental results in Sec. 5. We conclude with remarks in Sec. 6. Proofs and training details are deferred to Appendix.

## 2. Preliminaries and Related Work

In one-step generative modeling, we wish to align the generated sample distribution $q_\theta(\mathbf{x}) := \int \delta(\mathbf{x} - \mathbf{g}_\theta(\mathbf{z}))q(\mathbf{z})\,\mathrm{d}\mathbf{z}$ with the true data distribution $p(\mathbf{x})$. Here, $\mathbf{g}_\theta \colon \mathcal{Z} \to \mathcal{X}$ is a parametric neural sampler which is also often called an implicit generative model that transforms samples from a base measure $q(\mathbf{z})$. In this section, we review some popular methods for training generative models, which will serve as preliminaries for our framework. More detailed discussion on the literature is deferred to Appendix B.

**Generative Adversarial Networks.** The most prominent approach in training implicit generative models is the generative adversarial network (GAN) (Goodfellow et al., 2014). In its most standard and widely used form, it alternates between the gradient steps of discriminator and generator training, which are

$$\min_{\psi} \; \mathsf{E}_{p(\mathbf{x})}[\mathsf{sp}(-\ell_\psi(\mathbf{x}))] + \mathsf{E}_{q_\theta(\mathbf{x})}[\mathsf{sp}(\ell_\psi(\mathbf{x}))], \quad (1)$$

$$\min_{\theta} \; \mathsf{E}_{q_\theta(\mathbf{x})}[\mathsf{sp}(-\ell_\psi(\mathbf{x}))], \quad (2)$$

respectively, where $\mathsf{sp}(y) := \log(1 + e^y)$ denotes the softplus function.[1] Here, we will call $\ell_\psi(\mathbf{x})$ the *discriminator*, which is supposed to capture the *log density ratio* $\log \frac{p(\mathbf{x})}{q_\theta(\mathbf{x})}$.[2] This so-called adversarial training can be understood as min-

---

[1]The generator loss $\mathsf{E}_{q_\theta(\mathbf{x})}[\mathsf{sp}(-\ell_\psi(\mathbf{x}))]$ in the second line is the so-called *non-saturating* version, while the original GAN generator loss $\mathsf{E}_{q_\theta(\mathbf{x})}[-\mathsf{sp}(\ell_\psi(\mathbf{x}))]$ is referred to as *saturating*.

[2]Note the one-to-one correspondence between the standard definition of discriminator $D_\psi(\mathbf{x}) := \frac{\exp(\ell_\psi(\mathbf{x}))}{1+\exp(\ell_\psi(\mathbf{x}))} \in [0, 1]$.

imizing the Jensen–Shannon divergence (JSD) with the help of discriminator, via the variational characterization of JSD.

Despite the popularity of GANs, training them is notoriously difficult. Although various techniques have been proposed to regularize the GAN objective—through alternatives to JSD (Nowozin et al., 2016; Arjovsky et al., 2017; Mao et al., 2017), novel regularizers (Miyato et al., 2018), and specialized network architectures (Karras et al., 2021; Brock et al., 2019; Sauer et al., 2022)—the discriminator training remains unstable. This has sparked increasing interest in developing new objectives for training generative models which we briefly discuss below.

**Diffusion Models.** Diffusion models or score-based generative models (Sohl-Dickstein et al., 2015; Ho et al., 2020) are state-of-the-art generative models that are based on the principles of thermodynamic diffusion. Given a *forward stochastic differential equation (SDE) process*

$$\mathrm{d}\mathbf{x}_t = f(\mathbf{x}_t, t)\mathrm{d}t + g(t)\mathrm{d}\mathbf{w}_t,$$

where $f(\mathbf{x}_t, t)$ is the drift function, $g(t)$ is the diffusion function, and $\mathbf{w}_t$ represents a Brownian noise process, diffusion models simulate the *reverse (generative) process*, which is also an SDE

$$\mathrm{d}\mathbf{x}_t = [f(\mathbf{x}_t, t) - g(t)^2 \nabla_{\mathbf{x}_t} \log p(\mathbf{x}_t)]\mathrm{d}t + g(t)\mathrm{d}\bar{\mathbf{w}}_t.$$

An equivalent deterministic *probability flow ODE* with the same marginals as the SDE can also be used in practice:

$$\mathrm{d}\mathbf{x}_t = \left(f(\mathbf{x}_t, t) - \frac{1}{2}g(t)^2 \nabla_{\mathbf{x}_t} \log p(\mathbf{x}_t)\right)\mathrm{d}t.$$

Thus, to generate samples, diffusion models are trained to learn the score of the data distribution at multiple noise levels $\sigma_t$ via *denoising score matching (DSM)* (Vincent, 2011), i.e., by minimizing

$$\mathcal{L}_{\mathrm{DSM}}(\theta) = \mathsf{E}_{p(\mathbf{x})q(\mathbf{z})p(t)}\left[w(t)\|\mathbf{s}_\theta(\mathbf{x}_t; t) - \mathbf{s}(\mathbf{x}_t|\mathbf{x})\|^2\right],$$

where $p(t)$ denotes a distribution over different noise levels, $\mathbf{x}_t := \mathbf{x} + \sigma_t \boldsymbol{\epsilon}, \boldsymbol{\epsilon} \sim \mathcal{N}(0, \mathbf{I})$, and $\mathbf{s}(\mathbf{x}_t|\mathbf{x}) :=$

$\nabla_{\mathbf{x}_t} \log p(\mathbf{x}_t|\mathbf{x})$. It is easy to show that $\mathbf{s}_\theta(\mathbf{x}_t; t) = \nabla_{\mathbf{x}_t} \log p(\mathbf{x}_t)$ using Tweedie's formula (Robbins, 1956). Sampling can then be achieved by Langevin dynamics (Song & Ermon, 2019; Song et al., 2020) or via black-box ODE solvers (Karras et al., 2022b; Lu et al., 2022b;c).

**Diffusion Distillation.** In practical applications, running a diffusion model for multiple steps to generate a single sample can be prohibitively expensive. Distilling few-step generative models from a high-quality pretrained diffusion model has thus become popular (Luo et al., 2024a; Yin et al., 2024b;a; Salimans et al., 2024; Xie et al., 2024). To learn the generator's parameters, most, if not all, approaches aim to minimize the reverse Kullback–Leibler divergence (KLD) $D_{KL}(q_\theta \| p)$ averaged across multiple noise levels:

$$D_{KL}^{avg}(q_\theta \| p) := E_{q_\theta(\mathbf{x})p(t)q(\boldsymbol{\epsilon})}[\log q_\theta(\mathbf{x}_t) - \log p(\mathbf{x}_t)].$$

To update the parameters via gradient descent, the gradient of this divergence is computed as

$$\nabla_\theta D_{KL}^{avg}(q_\theta \| p) \tag{3}$$
$$= E_{q(\mathbf{z})p(t)q(\boldsymbol{\epsilon})}[\nabla_\theta \mathbf{g}_\theta(\mathbf{z})(\mathbf{s}_{q_\theta}(\mathbf{x}_t; t) - \mathbf{s}_p(\mathbf{x}_t; t)) |_{\mathbf{x}=\mathbf{g}_\theta(\mathbf{z})}],$$

where $\mathbf{s}_{q_\theta}$ and $\mathbf{s}_p$ are the noisy scores of the fake and true samples, respectively. In the distillation setup, a pretrained diffusion model is plugged in as a close proxy to the true noisy score $\mathbf{s}_p(\mathbf{x}_t; t)$, while the fake noisy score $\mathbf{s}_{q_\theta}(\mathbf{x}_t; t)$ is trained along with the generator to assist the training.

**Consistency Models.** Distillation approaches often rely on pretrained score models and may use expensive regularizers to address issues like mode collapse and improve sample quality (Yin et al., 2024b; Salimans et al., 2024). In contrast, consistency models (Song et al., 2023; Song & Dhariwal, 2024b), which can be trained from scratch, are trained to simulate the underlying probability flow ODE and ensure each sample along the trajectory maps to the origin. Consistency training, however, can be unstable and is known to sensitive to the noise schedule and distance function (Geng et al., 2025). Additionally, the architecture for consistency models need to be carefully chosen, as the approach relies on a single-sample approximation of Tweedie's formula, which is only valid when noise levels are closely spaced.

## 3. Training from Scratch

In this section, we introduce our new framework, *Score-of-Mixture Training* (SMT). We describe how to efficiently train one-step generative models from scratch, i.e., without a pretrained diffusion model. In Sec. 4, we explain how the framework can be adapted to leverage a pretrained diffusion model when available, referring to this variant as *Score-of-Mixture Distillation* (SMD).

The key ingredient of this framework is *distribution matching* using a new family of statistical divergences (Sec. 3.1),

whose gradient can be approximated by estimating the score of *mixture distributions* of real and fake distributions (Sec. 3.3), hence the name *Score of Mixture* Training. We adopt the concept of multi-noise level learning from diffusion models and propose multi-divergence minimization for stable training (Sec. 3.2). A practical implementation of our method is described in Sec. 3.4, followed by details of the training procedure in Sec. 3.5.

### 3.1. Minimizing $\alpha$-Skew Jensen–Shannon Divergences

The crux of the new framework lies in minimizing a class of statistical divergences between $p(\mathbf{x})$ and $q_\theta(\mathbf{x})$ defined as

$$D_{JSD}^{(\alpha)}(q_\theta, p) := \frac{1}{\alpha} D_{KL}(q_\theta \| \alpha p + (1 - \alpha)q_\theta)$$
$$+ \frac{1}{1 - \alpha} D_{KL}(p \| \alpha p + (1 - \alpha)q_\theta)$$

for some $\alpha \in (0, 1)$, which we call the $\alpha$-*skew Jensen-Shannon divergence* ($\alpha$-JSD) (Nielsen, 2010). This divergence belongs to $f$-divergences (Csiszár et al., 2004).

Interestingly, $\alpha$-skew JSD naturally interpolates between the forward Kullback–Leibler divergence (KLD) $D_{KL}(p \| q_\theta)$ (when $\alpha \to 0$), the standard definition of JSD (when $\alpha = \frac{1}{2}$), and the reverse KLD $D_{KL}(q_\theta \| p)$ (when $\alpha \to 1$). In contrast to the forward KLD and reverse KLD, the $\alpha$-skew JSD with $\alpha \in (0, 1)$ is well-defined even when there is a support mismatch in $p$ and $q_\theta$, which may be the case especially in the beginning of training.

> **Feature 1: Multi-Divergence Training.** Hence, we propose to minimize a weighted sum of the $\alpha$-JSD's for different $\alpha$'s, as divergences with different $\alpha$'s exploit different geometries between two distributions. For example, it is known that minimizing the forward and reverse KLD leads to mode-covering and mode-seeking behaviors, respectively, and we can enforce better support matching behavior by considering the entire range of $\alpha$.

To minimize this family of divergences in practice, we consider its gradient expression:

**Proposition 3.1.** *Suppose that* $E_{q_\theta(\mathbf{x})}[\nabla_\theta \log q_\theta(\mathbf{x})] = 0.$[3] *Then, we have*

$$\nabla_\theta D_{JSD}^{(\alpha)}(q_\theta, p) \tag{4}$$
$$= \frac{1}{\alpha} E_{q(\mathbf{z})}\left[\nabla_\theta \mathbf{g}_\theta(\mathbf{z})(\mathbf{s}_{\theta;0}(\mathbf{x}) - \mathbf{s}_{\theta;\alpha}(\mathbf{x}))\Big|_{\mathbf{x}=\mathbf{g}_\theta(\mathbf{z})}\right],$$

*where we define the score of the mixture distribution*

$$\mathbf{s}_{\theta;\alpha}(\mathbf{x}) := \nabla_\mathbf{x} \log(\alpha p(\mathbf{x}) + (1 - \alpha)q_\theta(\mathbf{x})).$$

---

[3]It is a standard assumption in the literature (Hyvärinen, 2005), which holds under a mild regularity assumption on the parametric model $q_\theta(\mathbf{x})$ so that $\int \nabla_\theta q_\theta(\mathbf{x}) \, d\mathbf{x} = \nabla_\theta \int q_\theta(\mathbf{x}) \, d\mathbf{x}$.

This proposition suggests that we can update the generator $\mathbf{g}_\theta(\mathbf{z})$ using this gradient expression, provided that we can estimate the *score of the mixture distribution* $\mathbf{s}_{\theta;\alpha}(\mathbf{x})$.

**Feature 2: Amortized Score Model.** To implement this idea, in this paper, we propose to use an *amortized score model* $(\mathbf{x}, \alpha) \mapsto \mathbf{s}_\psi(\mathbf{x}; \alpha)$, to approximate the score of mixture $\mathbf{s}_{\theta;\alpha}(\mathbf{x})$. Through our experiments we show that learning the scores of mixture over different $\alpha$'s using a single model is effective and helps training. In Sec. 3.3, we explain how we can train the amortized score model $(\mathbf{x}, \alpha) \mapsto \mathbf{s}_\psi(\mathbf{x}; \alpha)$ using samples from $p(\mathbf{x})$ and $q_\theta(\mathbf{x})$.

### 3.2. Learning with Multiple Noise Levels

To achieve stable training, we opt to minimize the divergence at different noise levels by considering the convolved distributions, $p_t := p * \mathcal{N}(\mathbf{0}, \sigma_t^2 \mathbf{I}_D)$ and $q_{\theta,t} := q_\theta * \mathcal{N}(\mathbf{0}, \sigma_t^2 \mathbf{I}_D)$. This idea is widely used in the existing distillation methods. We borrow the variance-exploding Gaussian noising process notation from Karras et al. (2022b) where $\sigma_t \in [\sigma_{\min}, \sigma_{\max}]$. As we also integrate over different $\alpha$'s, the final objective becomes

$$\mathcal{L}_{\text{gen}}(\theta) := \mathsf{E}_{p(\alpha)p(t)}[\mathrm{D}_{\text{JSD}}^{(\alpha)}(q_{\theta,t}, p_t)], \qquad (5)$$

where we will prescribe the choice of $p(\alpha)$ in Sec. 3.5. Similar to Eq. (4), the gradient of the divergence at noise level $t$ can be approximated via the amortized score as

$$\nabla_\theta \mathrm{D}_{\text{JSD}}^{(\alpha)}(q_{\theta,t}, p_t) \approx \boldsymbol{\gamma}_\psi(\theta; \alpha, t) \qquad (6)$$
$$:= \mathsf{E}_{q(\mathbf{z})q(\boldsymbol{\epsilon})}\left[\nabla_\theta \mathbf{g}_\theta(\mathbf{z}) \frac{\mathbf{s}_\psi(\mathbf{x}_t; 0, t) - \mathbf{s}_\psi(\mathbf{x}_t; \alpha, t)}{\alpha}\bigg|_{\mathbf{x}=\mathbf{g}_\theta(\mathbf{z})}\right],$$

where the amortized score model $\mathbf{s}_\psi(\mathbf{x}_t; \alpha, t)$, which is conditioned on the noise level $t$, is an estimate of $\mathbf{s}_{\theta;\alpha,t}(\mathbf{x}_t) := \nabla_{\mathbf{x}_t} \log(\alpha p(\mathbf{x}_t) + (1 - \alpha)q_\theta(\mathbf{x}_t))$. We provide a practical implementation of the amortized score model as a small modification of a diffusion model architecture in Sec. 3.4. We remark in passing that this expression can be understood as a generalization of the gradient update of Eq. (3) used in the existing reverse-KLD-based distillation schemes.

Finally, we can then approximate the generator gradient as

$$\nabla_\theta \mathcal{L}_{\text{gen}}(\theta) \approx \mathsf{E}_{p(\alpha)p(t)}[\boldsymbol{\gamma}_\psi(\theta; \alpha, t)].$$

Importantly, similar to existing distillation methods, the gradient only involves the output of the score model, but not its gradient. This is beneficial since such extra gradient information requires expensive backpropagation through the score model to the generator (Zhou et al., 2024).

### 3.3. Estimating Score of Mixture Distributions

Estimating the score of the mixture distribution turns out to be as simple as minimizing a mixture of the score matching losses, as stated in the following proposition:

**Proposition 3.2.** *For any $\alpha \in [0, 1]$, the minimizer of the objective function*

$$\mathcal{L}(\psi; \alpha) = \alpha \, \mathsf{E}_{p(\mathbf{x})}[\|\mathbf{s}_\psi(\mathbf{x}; \alpha) - \mathbf{s}_p(\mathbf{x})\|^2]$$
$$+ (1 - \alpha) \, \mathsf{E}_{q_\theta(\mathbf{x})}[\|\mathbf{s}_\psi(\mathbf{x}; \alpha) - \mathbf{s}_{q_\theta}(\mathbf{x})\|^2] \qquad (7)$$

*satisfies* $\mathbf{s}_{\psi^*}(\mathbf{x}; \alpha) = \mathbf{s}_{\theta;\alpha}(\mathbf{x})$.

Since we train with multiple noise levels, we are interested in the marginal score of $\mathbf{x}_t = \mathbf{x} + \sigma_t \boldsymbol{\epsilon}, \boldsymbol{\epsilon} \sim \mathcal{N}(0, \mathbf{I})$ at some noise level $\sigma_t$. We can use denoising score matching (Vincent, 2011) to define an equivalent *sample-only* objective to learn the score using Tweedie's formula. Namely, to approximate $\mathbf{s}_{\theta;\alpha,t}(\mathbf{x})$ using the amortized score model $\mathbf{s}_\psi(\mathbf{x}; \alpha, t)$, we can minimize

$$\mathcal{L}_{\text{score}}(\psi) := \mathsf{E}_{p(\alpha)p(t)}[\mathcal{L}_{\text{score}}(\psi; \alpha, t)],$$

where

$$\mathcal{L}_{\text{score}}(\psi; \alpha, t) \qquad (8)$$
$$:= \alpha \, \mathsf{E}_{p(\mathbf{x})q(\boldsymbol{\epsilon})}[\|\mathbf{s}_\psi(\mathbf{x}_t; \alpha, t) + \boldsymbol{\epsilon}/\sigma_t\|^2]$$
$$+ (1 - \alpha) \, \mathsf{E}_{q_\theta(\mathbf{x})q(\boldsymbol{\epsilon})}[\|\mathbf{s}_\psi(\mathbf{x}_t; \alpha, t) + \boldsymbol{\epsilon}/\sigma_t\|^2].$$

See Proposition A.1 for a formal statement. In practice, we parametrize the score model in the form of a *denoiser* and reconstruct the score from the denoiser output via Tweedie's formula; see Appendix C.1.

**Feature 3: Leveraging Real and Fake Samples.** We remark that our score learning objective seamlessly utilizes both real and fake samples throughout the training, helping the generator better generalize. This is in contrast to some existing diffusion distillation methods, which introduce expensive regularizers to integrate real samples, or backpropagate through the pretrained score model (Yin et al., 2024b;a; Salimans et al., 2024).

### 3.4. Practical Design of Amortized Score Network

With an additional conditioning scheme to embed auxiliary information about $\alpha$ in addition to the noise level $\sigma_t$, any existing diffusion model backbone can be used to parameterize the amortized score network $\mathbf{s}_\psi(\mathbf{x}; \alpha, t)$. Here, we describe how we can modify the popular UNet-based score architectures (Song et al., 2020; Nichol & Dhariwal, 2021; Karras et al., 2022b) with minimal modifications.

First, drawing from the noise embedding sensitivity analysis by Song & Dhariwal (2024b), we opt for a Fourier embedding $\mathbf{c}_\alpha$ with a default scale of 16. This choice ensures that the embedding is sufficiently sensitive to fluctuations in $\alpha$, particularly during the early stage of training.

Then, we concatenate the $\alpha$-embedding with the embedding of other auxiliary information (e.g., $t$ and labels) and apply

a single SiLU (Elfwing et al., 2018) activated linear layer:

$$\mathbf{c}_{\text{out}} = \text{silu}(\mathbf{W}_{\text{aux}}\mathbf{c}_{\text{aux}} + \mathbf{W}_\alpha\mathbf{c}_\alpha).$$

The rationale behind this choice is as follows: as training progresses, the real and fake distributions begin to overlap, making it natural for the amortized score model to become less sensitive to $\alpha$. Thanks to the additional linear layer $\mathbf{W}_\alpha$ after the $\alpha$-embedding $\mathbf{c}_\alpha$, this behavior can be realized when $\mathbf{W}_\alpha \approx \mathbf{0}$, when necessary.

### 3.5. Training

**Alternating Training.** Our training scheme alternates between the score estimation with the score matching objective in Eq. (8), and the generator training with Eq. (6), where we plug-in $\mathbf{s}_\psi(\mathbf{x}_t; \alpha, t)$ in place of $\mathbf{s}_{\theta;\alpha,t}(\mathbf{x}_t)$. This is similar in spirit to GAN training, but the DSM technique in our framework in place of the discriminator training naturally stabilizes training. The overall training framework is summarized in Fig. 1 and Alg. 1 in Appendix C.

**Initialization.** We warm up the generator with a standard denoising task as in diffusion models for several steps to better initialize the weights, as we empirically found that initializing the generator with pretrained weights from a denoiser significantly accelerated convergence. The amortized score network is randomly initialized.

**Choice of $p(\alpha)$.** The choice of $p(\alpha)$ is crucial in our framework. To train both the generator and score model, we sample $\alpha$ from a uniform distribution over 1000 equally spaced points in $[0, 1]$, ensuring a dense enough grid to generalize to any $\alpha$. For score training, we further ensure that 25% of the sampled $\alpha$'s are zero, since this is always used in our gradient update; see Eq. (6).

**Adaptive Weighting.** In practice we compute the gradient with an adaptive weight $w(\mathbf{x}_t, \mathbf{x}, \alpha, t)$ to ensure that the scale of the gradient for each minibatch sample is roughly uniform for different values of $\alpha$ and $t$. Hence, we modify the generator gradient in Eq. (6) as

$$\gamma_\psi^w(\theta; \alpha, t) := \mathsf{E}_{q(\mathbf{z})}\Big[\nabla_\theta\mathbf{g}_\theta(\mathbf{z})\times \tag{9}$$
$$\Big\{w(\mathbf{x}_t, \mathbf{x}, \alpha, t)\frac{\mathbf{s}_\psi(\mathbf{x}_t; 0, t) - \mathbf{s}_\psi(\mathbf{x}_t; \alpha, t)}{\alpha}\Big\}\Big|_{\mathbf{x}=\mathbf{g}_\theta(\mathbf{z})}\Big],$$

where the weighting is defined as

$$w(\mathbf{x}_t, \mathbf{x}, \alpha, t) := w_\alpha(\mathbf{x}_t, t)w_{\text{DMD}}(\mathbf{x}_t, \mathbf{x}, t). \tag{10}$$

Here $w_{\text{DMD}}$ is the adaptive noise weighting introduced by (Yin et al., 2024b) (see Eq. (32) in Appendix B) and $w_\alpha(\mathbf{x}_t, t)$ is a new weighting inspired by the pseudo-Huber norm (Song & Dhariwal, 2024a; Geng et al., 2025)

$$w_\alpha(\mathbf{x}_t, t) := \alpha\sqrt{\frac{\|\mathbf{s}_\psi(\mathbf{x}_t; 0, t) - \mathbf{s}_\psi(\mathbf{x}_t; 1, t)\|^2}{\|\mathbf{s}_\psi(\mathbf{x}_t; 0, t) - \mathbf{s}_\psi(\mathbf{x}_t; \alpha, t)\|^2}}.$$

This weighting still preserves the limiting forward KLD behavior of the objective as $\alpha \to 0$ and simplifies to DMD gradient when $\alpha = 1$. We empirically show the efficacy of our adaptive weighting term $w_\alpha(\mathbf{x}_t, t)$ through ablation studies on the CIFAR-10 dataset in Sec. 5.3; see Fig. 2b.

**Regularization with GAN.** We empirically found that a GAN-type regularization can accelerate convergence even further in the beginning of training. More concretely, we can train the discriminator $\ell_\psi(\mathbf{x}_t; t) \approx \log\frac{p(\mathbf{x})}{q_\theta(\mathbf{x})}$ by the GAN discriminator training in Eq. (1). In our implementation, we opt to train a discriminator using a variant based on the $\alpha$-JSD, as described in Appendix D.2. Given a discriminator $\ell_\psi(\mathbf{x}_t; t)$, we minimize a *non-saturating version* of the $\alpha$-JSD loss (cf. Eq. (2)),

$$\mathcal{L}_{\text{GAN}}^{(\alpha,t)}(\theta) = \mathsf{E}_{q_\theta(\mathbf{x}_t)}\Big[\text{sp}\Big(-\ell_\psi(\mathbf{x}_t; t) - \log\frac{\alpha}{1-\alpha}\Big)\Big]. \tag{11}$$

The derivation can be found in Appendix D.2. Similar to Yin et al. (2024a), we parameterized the discriminator by a stack of convolution layers, applied on top of an intermediate feature of the amortized score network at $\alpha = 1/2$.

Similar to DMD2 (Yin et al., 2024a), we implement a GAN discriminator building on top of the score network, with only a few additional MLP layers. This score-model-dependent design allows the full model to benefit from the training stability provided by denoising score matching, while the GAN discriminator loss only trains the small auxiliary MLP. (For ImageNet, the generator has 296M parameters and the discriminator has 18M.) Thus, the discriminator represents a small fraction of the overall model size and has a negligible impact on training speed. As a result, our use of the GAN regularizer is both efficient and stable.

## 4. Distilling from Pretrained Diffusion Model

In our development so far, we do not assume access to a pretrained diffusion model. In this section, we show how a practitioner can train a one-step generative model leveraging a pretrained diffusion model, if available, within our framework. The proposed distillation scheme is comparable or even outperforms the state-of-the-art distillation schemes.

### 4.1. How To Leverage Pretrained Diffusion Model

In the distillation setup, we treat the pretrained diffusion model as the data score $\mathbf{s}_p(\mathbf{x}_t; t)$, and thus training the score of mixture $\mathbf{s}_{\theta;\alpha}(\mathbf{x}_t; t)$ using a single, amortized model may not be the most efficient parameterization. Hence, instead, we consider the following expression

$$\mathbf{s}_{\theta;\alpha}(\mathbf{x}) = D_{\theta;\alpha}(\mathbf{x})\mathbf{s}_p(\mathbf{x}) + (1 - D_{\theta;\alpha}(\mathbf{x}))\mathbf{s}_{q_\theta}(\mathbf{x}),$$

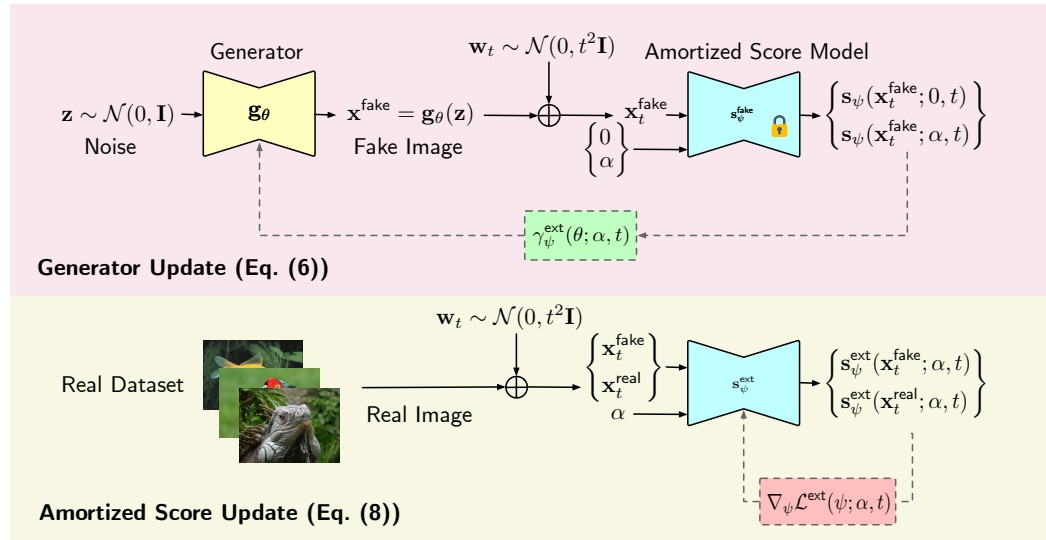

*Figure 1.* Overview of SMT. **Top:** To update the generator, we compute the gradient of the $\alpha$-JSD on noisy fake samples with the *frozen* amortized score model using Eq. (6). **Bottom:** The amortized score model is updated by computing the score of the mixture distribution on both fake and real noisy samples, and then updating the weights using the gradient in Eq. (8).

where

$$D_{\theta;\alpha}(\mathbf{x}) := \frac{\alpha p(\mathbf{x})}{\alpha p(\mathbf{x}) + (1-\alpha)q_\theta(\mathbf{x})}$$

$$= \sigma\left(\log\frac{p(\mathbf{x})}{q_\theta(\mathbf{x})} + \log\frac{\alpha}{1-\alpha}\right).$$

See Proposition A.2 for a formal statement. In words, we can express the score of mixture $\mathbf{s}_{\theta;\alpha}(\mathbf{x})$ as a mixture of scores $\mathbf{s}_p$ and $\mathbf{s}_{q_\theta}$, where the weight is $(D_{\theta;\alpha}(\mathbf{x}), 1 - D_{\theta;\alpha}(\mathbf{x}))$. This suggests that instead of an amortized modeling of the score of mixture, we can use an alternative parameterization,

$$\mathbf{s}_\psi^{\mathsf{exp}}(\mathbf{x};\alpha) := D_\psi(\mathbf{x};\alpha)\mathbf{s}_p(\mathbf{x}) + (1 - D_\psi(\mathbf{x};\alpha))\mathbf{s}_\psi^{\mathsf{fake}}(\mathbf{x}),$$

where

$$D_\psi(\mathbf{x};\alpha) := \sigma\left(\ell_\psi(\mathbf{x}) + \log\frac{\alpha}{1-\alpha}\right).$$

Here, we can parameterize the *discriminator* $\mathbf{x} \mapsto \ell_\psi(\mathbf{x})$ in the same way as we do for the GAN discriminator.

We can extend this to multiple noise levels easily. Hence, an alternative parameterization for $\mathbf{s}_{\theta;\alpha}(\mathbf{x}_t; t)$ is

$$\mathbf{s}_\psi^{\mathsf{exp}}(\mathbf{x}_t;\alpha,t) := D_\psi(\mathbf{x}_t;\alpha,t)\mathbf{s}_p(\mathbf{x}_t;t) \quad (12)$$
$$+ (1 - D_\psi(\mathbf{x}_t;\alpha,t))\mathbf{s}_\psi^{\mathsf{fake}}(\mathbf{x}_t;t),$$

where

$$D_\psi(\mathbf{x}_t;\alpha,t) := \sigma\left(\ell_\psi(\mathbf{x}_t;t) + \log\frac{\alpha}{1-\alpha}\right). \quad (13)$$

Plugging this explicit score model into Eq. (8), we can learn both the fake score model $\mathbf{s}_\psi^{\mathsf{fake}}$ and the discriminator $\ell_\psi$ at different noise levels.

**Corollary 4.1.** *Let $\alpha \in [0, 1]$ be fixed and $\sigma_t$ be some fixed noise level. Then, the minimizer of the objective function*

$$\mathcal{L}^{\mathsf{exp}}(\psi;\alpha,t) \quad (14)$$
$$:= \alpha\, \mathsf{E}_{p(\mathbf{x})q(\boldsymbol{\epsilon})}[\|\mathbf{s}_\psi^{\mathsf{exp}}(\mathbf{x}_t;\alpha,t) + \boldsymbol{\epsilon}/\sigma_t\|^2]$$
$$+ (1-\alpha)\, \mathsf{E}_{q_\theta(\mathbf{x})q(\boldsymbol{\epsilon})}[\|\mathbf{s}_\psi^{\mathsf{exp}}(\mathbf{x}_t;\alpha,t) + \boldsymbol{\epsilon}/\sigma_t\|^2]$$

*satisfies*

$$\mathbf{s}_{\psi^*}^{\mathsf{fake}}(\mathbf{x};t) = \mathbf{s}_{q_\theta}(\mathbf{x};t) \ \ and \ \ \ell_{\psi^*}(\mathbf{x}_t;t) = \log\frac{p(\mathbf{x}_t)}{q_\theta(\mathbf{x}_t)}.$$

We remark that this new regression objective in Eq. (14) provides a new way to compute the log density ratio, as an alternative to the GAN training (see Eq. (1)). In Appendix D.3, we establish a connection between this objective for training a discriminator to an existing GAN discriminator objective in the literature.

With this new, explicit parameterization, we can approximate the gradient expression in Eq. (6) as

$$\nabla_\theta \mathsf{D}_{\mathsf{JSD}}^{(\alpha)}(q_{\theta,t}, p_t)$$
$$\approx \boldsymbol{\gamma}_\psi^{\mathsf{exp}}(\theta;\alpha,t) \quad (15)$$
$$:= \mathsf{E}_{q(\mathbf{z})}\Big[D_\psi(\mathbf{x}_t;\alpha,t)\times$$
$$\nabla_\theta\mathbf{g}_\theta(\mathbf{z})\frac{\mathbf{s}_\psi^{\mathsf{fake}}(\mathbf{x}_t;t) - \mathbf{s}_p(\mathbf{x}_t;t)}{\alpha}\Big|_{\mathbf{x}=\mathbf{g}_\theta(\mathbf{z})}\Big].$$

### 4.2. Implementation and Training

**Model Architectures.** We can leverage any existing diffusion model architectures directly for the fake score

$\mathbf{s}_\psi^{\text{fake}}(\mathbf{x}_t; t)$. We parametrize the discriminator $\ell_\psi(\mathbf{x}_t; t)$ similar to the noise-conditional discriminator in our training from scratch setting (see Sec. 3.5). The difference is that we can train the discriminator by minimizing the DSM loss in Eq. (14) naturally, without an additional GAN loss. When training the generator, we plug in this approximate log density ratio into Eq. (11) to regularize the generator updates.

**Training.** We also train in an alternating fashion. Since we have access to a pretrained score model, we use this to initialize the weights of both the generator and the fake score model. We utilize the same sampling distribution for $\alpha$ as in our training from scratch setup (see Sec. 3.5). The procedure is summarized in Fig. 4 and Alg. 2 in Appendix C.

# 5. Experiments

In this section, we first present results on the ImageNet $64 \times 64$ dataset. We then demonstrate the competitiveness of our method on the CIFAR-10 dataset and conduct a series of ablation studies. We measure performance through sample quality as measured by the Fréchet Inception Distance (FID) (Heusel et al., 2017). The exact hyperparameters, training configurations used and additional results, including an example training dynamics and latent interpolation, can be found in Appendix E. Our implementation can be found at `https://github.com/tkj516/score-of-mixture-training`.

## 5.1. Class-conditional ImageNet 64x64 Generation

**Experimental Setup.** We trained class-conditional one-step generative models on ImageNet $64 \times 64$ (Deng et al., 2009), experimenting with both distillation and training from scratch. **In both cases**, we used the ADM architecture (Nichol & Dhariwal, 2021) as the base score model architecture, and the discriminator $\ell_\psi(\mathbf{x}_t; t)$ was implemented as a stack of convolution layers operating on the bottleneck feature from the score network, similar to DMD2 (Yin et al., 2024a). For **training from scratch**, we augmented the score architecture using an $\alpha$-embedding as described in Sec. 3.4. The total number of parameters of the amortized score model remained unchanged otherwise. As a warmup stage, we pretrained the generator on the dataset using a standard diffusion denoising objective for 40k steps to initialize the weights. For **distillation**, we used a pretrained diffusion model from (Karras et al., 2022b).

**Results.** We evaluated our method against several published baselines for both training from scratch and distillation. As shown in Table 2, when trained from scratch, our generator with 296M parameters outperforms both consistency training and its improved variant (Song et al., 2023; Song & Dhariwal, 2024a), with a much smaller training budget (200k iterations with batch size of 40 vs. 800k iterations

with batch size of 512). Our model also competes favorably with iCT-deep, despite using a generator with half the number of parameters: FID of 3.23 with 296M parameters (ours) vs. 3.25 with 592M parameters (iCT-deep). We observed stable training throughout, without requiring extensive hyperparameter tuning or special noise schedule adjustments as in consistency training, as visualized in Fig. 2a. We also surpass the ECT model (Geng et al., 2025) of similar size and training budget that includes several modifications to induce stability in consistency training. Samples generated using our method can be found in Fig. 3 and Appendix E.

In the distillation setting, our model achieves a competitive FID of 1.48, outperforming several baselines. Notably, we outperform consistency distillation methods, such as multistep consistency distillation (Heek et al., 2024), despite using only a fraction of the model size (256M parameters against 1200M parameters). Our model also surpasses consistency trajectory models (CTM) (Kim et al., 2024), which ensure consistency between random points along the PF ODE trajectory, by simulating the reverse diffusion sampler for an arbitrary number of steps per minibatch, thus resulting in high computational cost.

We also outperform reverse-KLD methods with similar compute or regularizers such as DMD (Yin et al., 2024b) and DMD2 (Yin et al., 2024a) with FIDs of 5.60 and 1.51 respectively. We note that on spending significant extra compute, DMD and DMD2 achieved improved results. For example, in the DMD framework without any GAN regularization, the authors simulate the reverse process of a diffusion model and sample several thousand noise-image pairs to anchor the generator's outputs. Each noise-image pair requires evaluating the diffusion denoiser 256 times for ImageNet 64×64, which is extremely costly in practice. In contrast in the DMD2 framework the authors adopt a lengthy finetuning stage with GAN regularization of 400k steps to further improve results.

We did not resort to any of the above techniques and sought to find an approach that worked best with a single execution of the training pipeline run for 200k steps.

## 5.2. Unconditional CIFAR-10 Generation

**Experimental Setup.** We evaluated our method on the CIFAR-10 dataset (Krizhevsky et al., 2009) for unconditional one-step generative modeling, considering both training from scratch and distillation. **In both cases**, we employed a DDPM++ architecture (Song et al., 2020) with EDM preconditioning (Karras et al., 2022b). The discriminator again followed the convolutional stack used in DMD2. For **training from scratch**, we modified the score model to incorporate the $\alpha$-embedding (Sec. 3.4) while maintaining a similar network size. To mitigate overfitting due to the dataset's small size, we enabled dropout with $p = 0.13$,

*Table 2.* Image generation results on ImageNet 64x64 (class-conditional) and CIFAR-10 32x32 (unconditional). The size of the sampler is denoted by the number of parameters (# params), and NFE stands for the Number of Function Evaluations. The best FIDs from each category are highlighted in bold, and our methods **SMT** and **SMD** are highlighted with a blue shade.

| Method | ImageNet 64x64 | | | CIFAR-10 32x32 | | |
|---|---|---|---|---|---|---|
| | # params | NFE | FID↓ | # params | NFE | FID↓ |
| *Training from scratch: Diffusion models* | | | | | | |
| DDPM (Ho et al., 2020) | - | - | - | 56M | 1000 | 3.17 |
| ADM (Dhariwal & Nichol, 2021) | 296M | 250 | 2.07 | - | - | - |
| EDM (Karras et al., 2022b) | 296M | 512 | **1.36** | 56M | 35 | **1.97** |
| *Training from scratch: One-step models* | | | | | | |
| 2-RF + distill (Liu et al., 2022) | - | - | - | 56M | 1 | 4.85 |
| CT (Song et al., 2023) | 296M | 1 | 13.0 | 56M | 1 | 8.70 |
| iCT (Song & Dhariwal, 2024a) | 296M | 1 | 4.02 | 56M | 1 | 2.83 |
| iCT-deep (Song & Dhariwal, 2024a) | 592M | 1 | 3.25 | 112M | 1 | **2.51** |
| ECT (Geng et al., 2025) | 280M | 1 | 5.51 | 56M | 1 | 3.60 |
| **SMT** (ours) | 296M | 1 | **3.23** | 56M | 1 | 3.13 |
| *Diffusion distillation* | | | | | | |
| PD (Salimans & Ho, 2022) | 296M | 1 | 10.7 | 60M | 1 | 9.12 |
| TRACT (Berthelot et al., 2023) | 296M | 1 | 7.43 | 56M | 1 | 3.78 |
| CD (LPIPS) (Song et al., 2023) | 296M | 1 | 6.20 | 56M | 1 | 4.53 |
| Diff-Instruct (Luo et al., 2024a) | 296M | 1 | 5.57 | 56M | 1 | 4.53 |
| MultiStep-CD (Heek et al., 2024) | 1200M | 1 | 3.20 | - | - | - |
| DMD w/o reg (Yin et al., 2024b) | 296M | 1 | 5.60 | 56M | 1 | 5.58 |
| DMD2 w/ GAN (Yin et al., 2024a) | 296M | 1 | 1.51 | 56M | 1 | 2.43 |
| MMD (Salimans et al., 2024) | 400M | 1 | 3.00 | - | - | - |
| SiD (Zhou et al., 2024) | 296M | 1 | 1.52 | 56M | 1 | **1.92** |
| SiM (Luo et al., 2024b) | - | - | - | 56M | 1 | 2.02 |
| 2-RF ++ (Lee et al., 2024) | 296M | 1 | 3.07 | 56M | 1 | 4.31 |
| **SMD** (ours) | 296M | 1 | **1.48** | 56M | 1 | 2.22 |
| *w/ expensive regularizer, simulation or finetuning* | | | | | | |
| CTM (Kim et al., 2024) | 296M | 1 | 1.92 | 56M | 1 | **1.98** |
| DMD w/ reg (Yin et al., 2024b) | 296M | 1 | 2.62 | 56M | 1 | 2.66 |
| DMD2 (finetuned) (Yin et al., 2024a) | 296M | 1 | **1.23** | - | - | - |

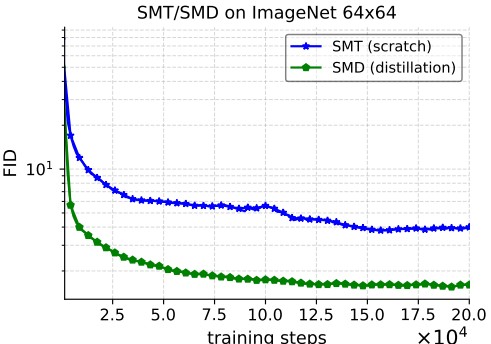

(a) ImageNet 64×64 (scratch and distillation).

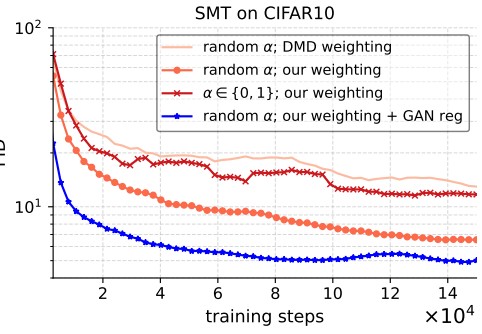

(b) CIFAR-10 with ablation studies (scratch).

*Figure 2.* FID evolution with training.

as in EDM. In the **distillation setting**, we initialized the generator with a pretrained unconditional diffusion model from (Karras et al., 2022b), using the same UNet backbone and weights. Distillation performed well without dropout.

**Results.** The last three columns in Table 2 highlight the performance of our method on CIFAR-10 compared to various baselines. In our training from scratch setting, despite utilizing a lower training budget (150k steps with a batch size of 40) than many methods, our approach remains highly competitive. In terms of training budget, the most comparable baseline is ECT, which we are able to outperform without requiring excessive design considerations and hyperparameter tuning. Our distillation results are also competitive. In particular, we outperform Diff-Instruct and DMD2, which are only based on minimizing the reverse KLD. This corroborates the benefit of our multi-divergence minimization approach. Image samples can be found in Appendix E.5.

### 5.3. Ablation Studies

We use the CIFAR-10 dataset to study the effectiveness of the design choices that we have proposed; see Fig. 2b.

**Choice of Adaptive Gradient Weighting.** Starting with our base objective without the GAN regularizer, we tested our $(\alpha, t)$-adaptive weighting in Eq. (10). Fig. 2b demonstrates the benefits of our weighting scheme, compared to the DMD weight function that only depends on $t$.

**Learning with Single vs. Multiple $\alpha$'s.** The $\alpha$-JSD reduces to the reverse KLD of DMD and other distillation methods, when $\alpha = 1$. To test the efficacy with multi-$\alpha$ learning, we implemented an amortized variant, training the score model only with $\alpha \in \{0, 1\}$. Results show that conditioning on a range of $\alpha$-values not only minimizes multiple divergences but also strengthens the $\alpha$ embedding as a conditioning signal thereby facilitating more accurate divergence minimization.

**Accelerated Convergence with GAN Regularizer.** We finally verify the benefits of our novel GAN-type regularizer for $\alpha$-JSD minimization. As demonstrated by the second and fourth curves in Fig. 2b, the GAN regularizer helps accelerate convergence especially in the beginning of training.

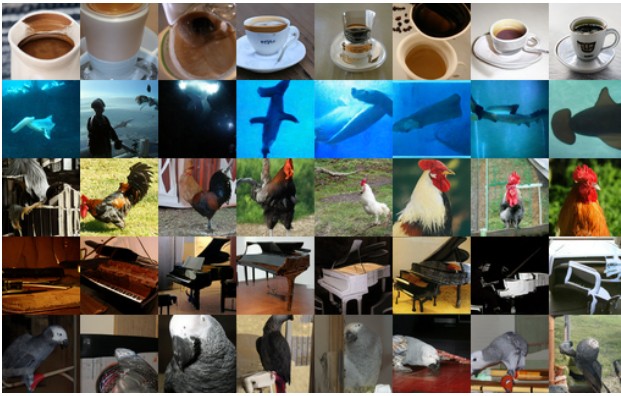

*Figure 3.* Samples from SMT on ImageNet 64×64. Each row represents a unique class. Additional samples can be found in Appendix E.5.

## 6. Concluding Remarks

In this paper, we show that high-quality one-step generative models can be trained from scratch and in a stable manner, without simulating the reverse diffusion process or probability flow ODE as in diffusion models and consistency models. The key distinctive idea in our framework is a new multi-divergence minimization paradigm implemented by estimating the score of mixture distributions. For stable training, we borrow multi-level noise learning and denoising score matching techniques from the diffusion literature. Our empirical results show that accurate score estimation facilitates stable minimization of statistical divergences. We hope this work offers a fresh perspective on generative modeling and inspires further research in the field.

**Limitations and Future Work.** While SMT/SMD achieve strong empirical performance, there is still room for improvement in both architecture and training strategies. Despite achieving highly competitive FID scores for one-step generation from scratch, models capable of few-step generation—such as consistency models—might further improve FID with additional iterations.Finally, given the generality of our framework, we believe these ideas could extend to other complex modalities, including speech and audio synthesis. We leave such directions for future work.

## Impact Statement

We introduce Score-of-Mixture Training, a simple yet effective one-step generative modeling framework that requires minimal design effort and hyperparameter tuning. We hope its ease of implementation will drive further research into efficient, state-of-the-art neural sampling. However, we acknowledge the potential risks of misuse, including the generation of fake, biased, or misleading content. Our work focuses on fundamental research using standard machine learning datasets, but we recognize the importance of ensuring generative models are secure and privacy-preserving to democratize this technology responsibly.

## Acknowledgements

This work was supported in part by the MIT-IBM Watson AI Lab under Agreement No. W1771646, by MIT Lincoln Laboratory, by ONR under Grant No. N000014-23-1-2803, and by the Department of the Air Force Artificial Intelligence Accelerator under Cooperative Agreement No. FA8750-19-2-1000. The views and conclusions contained in this document are those of the authors and should not be interpreted as representing the official policies, either expressed or implied, of the Department of the Air Force or the U.S. Government. The U.S. Government is authorized to reproduce and distribute reprints for Government purposes notwithstanding any copyright notation herein.

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

# Appendix

## A. Deferred Statements and Proofs

### A.1. Proof of Proposition 3.1

*Proof of Proposition 3.1.* We can simplify the gradient of each term separately as follows:

$$
\nabla_\theta \mathrm{D}_{\mathrm{KL}}(q_\theta \| \alpha p + (1-\alpha)q_\theta) = \mathsf{E}_{q_\theta(\mathbf{x})}\left[\nabla_\theta \log \frac{q_\theta(\mathbf{x})}{\alpha p(\mathbf{x}) + (1-\alpha)q_\theta(\mathbf{x})}\right] + \mathsf{E}_{q(\mathbf{z})}\left[\nabla_\theta \mathbf{g}_\theta(\mathbf{z})(\mathbf{s}_{\theta;0}(\mathbf{x}) - \mathbf{s}_{\theta;\alpha}(\mathbf{x}))\Big|_{\mathbf{x}=\mathbf{g}_\theta(\mathbf{z})}\right],
$$
$$
\nabla_\theta \mathrm{D}_{\mathrm{KL}}(p \| \alpha p + (1-\alpha)q_\theta) = -\mathsf{E}_{p(\mathbf{x})}\left[\nabla_\theta \log(\alpha p(\mathbf{x}) + (1-\alpha)q_\theta(\mathbf{x}))\right].
$$

Here, note that in the first expression, we invoke the chain rule: for some function $f_\theta \colon \mathcal{X} \to \mathbb{R}$, we have

$$
\nabla_\theta f_\theta(\mathbf{g}_\theta(\mathbf{z})) = (\nabla_\theta f_\theta(\mathbf{x}))|_{\mathbf{x}=\mathbf{g}_\theta(\mathbf{z})} + \nabla_\theta \mathbf{g}_\theta(\mathbf{z})(\nabla_{\mathbf{x}} f_\theta(\mathbf{x}))|_{\mathbf{x}=\mathbf{g}_\theta(\mathbf{z})}.
$$

Combining these two terms with the weights, we get the gradient of the $\alpha$-skew JSD:

$$
\begin{aligned}
\nabla_\theta \mathrm{D}_{\mathrm{JSD}}^{(\alpha)}(q_\theta, p) &= \frac{1}{\alpha}\nabla_\theta \mathrm{D}_{\mathrm{KL}}(q_\theta \| \alpha p + (1-\alpha)q_\theta) + \frac{1}{1-\alpha}\nabla_\theta \mathrm{D}_{\mathrm{KL}}(p \| \alpha p + (1-\alpha)q_\theta) \\
&= \frac{1}{\alpha}\mathsf{E}_{q(\mathbf{z})}\left[\nabla_\theta \mathbf{g}_\theta(\mathbf{z})(\mathbf{s}_{\theta;0}(\mathbf{x}) - \mathbf{s}_{\theta;\alpha}(\mathbf{x}))\Big|_{\mathbf{x}=\mathbf{g}_\theta(\mathbf{z})}\right] \\
&\quad - \frac{1}{\alpha(1-\alpha)}\mathsf{E}_{\alpha p(\mathbf{x})+(1-\alpha)q_\theta(\mathbf{x})}[\nabla_\theta \log(\alpha p(\mathbf{x}) + (1-\alpha)q_\theta(\mathbf{x}))] \\
&\quad + \frac{1}{\alpha}\mathsf{E}_{q_\theta(\mathbf{x})}[\nabla_\theta \log q_\theta(\mathbf{x})] \\
&= \frac{1}{\alpha}\mathsf{E}_{q(\mathbf{z})}\left[\nabla_\theta \mathbf{g}_\theta(\mathbf{z})(\mathbf{s}_{\theta;0}(\mathbf{x}) - \mathbf{s}_{\theta;\alpha}(\mathbf{x}))\Big|_{\mathbf{x}=\mathbf{g}_\theta(\mathbf{z})}\right].
\end{aligned}
$$

Here, we use the assumption that $\mathsf{E}_{q_\theta(\mathbf{x})}[\nabla_\theta \log q_\theta(\mathbf{x})] = 0$.      $\square$

## A.2. Proof of Proposition 3.2

*Proof of Proposition 3.2.* We can write the objective $\mathcal{L}(\psi; \alpha)$ as

$$\mathcal{L}(\psi; \alpha) = \int \left\{ (\alpha p(\mathbf{x}) + (1 - \alpha)q_\theta(\mathbf{x})) \|\mathbf{s}_\psi(\mathbf{x}; \alpha)\|^2 - 2(\alpha p(\mathbf{x})s_p(\mathbf{x}) + (1 - \alpha)q_\theta(\mathbf{x}))^\mathsf{T}\mathbf{s}_\psi(\mathbf{x}; \alpha) \right\} d\mathbf{x} + C$$

$$= \int (\alpha p(\mathbf{x}) + (1 - \alpha)q_\theta(\mathbf{x})) \left\| \mathbf{s}_\psi(\mathbf{x}; \alpha) - \frac{\alpha p(\mathbf{x})\mathbf{s}_p(\mathbf{x}) + (1 - \alpha)q_\theta(\mathbf{x})\mathbf{s}_{q_\theta}(\mathbf{x})}{\alpha p(\mathbf{x}) + (1 - \alpha)q_\theta(\mathbf{x})} \right\|^2 d\mathbf{x} + C'.$$

Hence, it is clear that the global minimizer should be

$$\mathbf{s}_{\psi^*}(\mathbf{x}; \alpha) = \frac{\alpha p(\mathbf{x})\mathbf{s}_p(\mathbf{x}) + (1 - \alpha)q_\theta(\mathbf{x})\mathbf{s}_{q_\theta}(\mathbf{x})}{\alpha p(\mathbf{x}) + (1 - \alpha)q_\theta(\mathbf{x})}$$

$$= \frac{\alpha \nabla_\mathbf{x} p(\mathbf{x}) + (1 - \alpha)\nabla_\mathbf{x} q_\theta(\mathbf{x})}{\alpha p(\mathbf{x}) + (1 - \alpha)q_\theta(\mathbf{x})}$$

$$= \frac{\nabla_\mathbf{x}(\alpha p(\mathbf{x}) + (1 - \alpha)q_\theta(\mathbf{x}))}{\alpha p(\mathbf{x}) + (1 - \alpha)q_\theta(\mathbf{x})}$$

$$= \nabla_\mathbf{x} \log(\alpha p(\mathbf{x}) + (1 - \alpha)q_\theta(\mathbf{x})). \qquad \square$$

## A.3. Deferred Statements

**Proposition A.1.** *Let $\alpha \in [0, 1]$ be fixed and $\sigma_t$ be some fixed noise level. Then, the minimizer of the objective function*

$$\mathcal{L}_{score}(\psi; \alpha, t) := \alpha \; \mathsf{E}_{p(\mathbf{x})q(\boldsymbol{\epsilon})}[\|\mathbf{s}_\psi(\mathbf{x}_t; \alpha, t) + \boldsymbol{\epsilon}/\sigma_t\|^2] + (1 - \alpha) \; \mathsf{E}_{q_\theta(\mathbf{x})q(\boldsymbol{\epsilon})}[\|\mathbf{s}_\psi(\mathbf{x}_t; \alpha, t) + \boldsymbol{\epsilon}/\sigma_t\|^2] \qquad (16)$$

*satisfies*

$$\mathbf{s}_{\psi^*}(\mathbf{x}_t; \alpha, t) = \mathbf{s}_{\theta;\alpha,t}(\mathbf{x}_t).$$

*Proof.* We can write the objective $\mathcal{L}(\psi; \alpha, t)$ as

$$\mathcal{L}_{score}(\psi; \alpha, t) = \iint (\alpha p(\mathbf{x}_t) + (1 - \alpha)q_\theta(\mathbf{x}_t)) \left\| \mathbf{s}_\psi(\mathbf{x}_t; \alpha, t) + \frac{\boldsymbol{\epsilon}}{\sigma_t} \right\|^2 d\mathbf{x} \, d\boldsymbol{\epsilon}.$$

This is a standard minimum mean square estimation (MMSE) problem for which the global minimizer is the conditional mean,

$$\mathbf{s}_{\psi^*}(\mathbf{x}_t; \alpha, t) = -\frac{1}{\sigma_t} \mathsf{E}_{\alpha p_t + (1-\alpha)q_{\theta,t}}[\boldsymbol{\epsilon}|\mathbf{x}_t]$$

$$= -\frac{1}{\sigma_t^2} \mathsf{E}_{\alpha p_t + (1-\alpha)q_{\theta,t}}[\mathbf{x}_t - \mathbf{x}|\mathbf{x}_t]$$

$$= -\frac{1}{\sigma_t^2}\mathbf{x}_t + \frac{1}{\sigma_t^2} \mathsf{E}_{\alpha p_t + (1-\alpha)q_{\theta,t}}[\mathbf{x}|\mathbf{x}_t]$$

$$= \nabla_{\mathbf{x}_t} \log(\alpha p(\mathbf{x}_t) + (1 - \alpha)q_\theta(\mathbf{x}_t)).$$

Here we use that $\mathbf{x}_t = \mathbf{x} + \sigma_t \boldsymbol{\epsilon}$ and make the connection to the marginal score in the last line using Tweedie's formula (Robbins, 1956). $\qquad \square$

**Proposition A.2.** *Let $\alpha \in [0, 1]$, $\mathbf{s}_p(\mathbf{x})$ be the data score, $\mathbf{s}_{q_\theta}(\mathbf{x})$ be the score of the generated samples. Then, the score of the mixture distribution can be expressed as*

$$\mathbf{s}_{\theta;\alpha}(\mathbf{x}) = D_{\theta;\alpha}(\mathbf{x})\mathbf{s}_p(\mathbf{x}) + (1 - D_{\theta;\alpha}(\mathbf{x}))\mathbf{s}_{q_\theta}(\mathbf{x}), \qquad (17)$$

*where*

$$D_{\theta;\alpha}(\mathbf{x}) := \sigma\left(\log \frac{p(\mathbf{x})}{q_\theta(\mathbf{x})} + \log \frac{\alpha}{1 - \alpha}\right), \qquad (18)$$

*Proof.* The amortized score can be expressed as

$$\nabla_{\mathbf{x}} \log(\alpha p(\mathbf{x}) + (1-\alpha)q_\theta(\mathbf{x})) = \frac{\nabla_{\mathbf{x}}(\alpha p(\mathbf{x}) + (1-\alpha)q_\theta(\mathbf{x}))}{\alpha p(\mathbf{x}) + (1-\alpha)q_\theta(\mathbf{x})}$$

$$= \frac{\alpha p(\mathbf{x})}{\alpha p(\mathbf{x}) + (1-\alpha)q_\theta(\mathbf{x})}\nabla_{\mathbf{x}} \log p(\mathbf{x}) + \frac{(1-\alpha)q_\theta(\mathbf{x})}{\alpha p(\mathbf{x}) + (1-\alpha)q_\theta(\mathbf{x})}\nabla_{\mathbf{x}} \log q_\theta(\mathbf{x})$$

$$= D(\mathbf{x};\alpha)\nabla_{\mathbf{x}} \log p(\mathbf{x}) + (1 - D(\mathbf{x};\alpha))\nabla_{\mathbf{x}} \log q_\theta(\mathbf{x}).$$

We can now simplify the scaling factor as

$$D(\mathbf{x};\alpha) = \frac{\alpha p(\mathbf{x})}{\alpha p(\mathbf{x}) + (1-\alpha)q_\theta(\mathbf{x})} = \sigma\left(\log\frac{p(\mathbf{x})}{q_\theta(\mathbf{x})} + \log\frac{\alpha}{1-\alpha}\right). \qquad\square$$

## B. Detailed Discussions on Related Work

### B.1. Diffusion Models

**Prior Work.** Sohl-Dickstein et al. (2015) first introduced diffusion probabilistic models (DPMs) as deep variational autoencoders (Kingma, 2014) based on the principles of thermodynamic diffusion with a Markov-chain variational posterior that maximizes the evidence lower bound (ELBO). Several years later, Ho et al. (2020) re-introduced DPMs (DDPMs) with modern neural network architectures and a simplified loss function that set a new state-of-the-art in image generation. Since then, numerous connections to existing literature in statistics, information theory and stochastic differential equations (SDEs) have helped bolster the quality of these models. For example, Song & Ermon (2019) illustrate the equivalence between DDPMs and DSM at multiple noise levels, thus bridging the areas of diffusion-based models and score-based models. Subsequently, Song et al. (2021b) showed that in continuous time, DPMs can be appropriately interpreted as solving for the reverse of a noising process that evolves as an SDE while Kingma et al. (2021) demonstrated that continuous-time DPMs can interpreted as VAEs and that the variational lower bound is invariant to the noise schedule except for its endpoints, thus bolstering its density estimation capabilities. Following the latter discovery, Kong et al. (2023) show that DPMs can in-fact be used for *exact* likelihood computation by leveraging techniques from information theory. To further improve DPMs, extensive research has gone into the choice of noise schedules, network architectures and loss functions (Nichol & Dhariwal, 2021; Hoogeboom et al., 2023; Karras et al., 2022a; Kingma & Gao, 2024). Many tangentially discovered frameworks such as rectified flows (Liu et al., 2023) and conditional normalizing flows trained with Gaussian conditional flow matching (Lipman et al., 2023), are also particular instances of (Gaussian) diffusion models with specialized noise schedules and weighted loss functions, as show in (Kingma & Gao, 2024).

**Formulation.** We take the following unified view in our definition of DPMs as inspired by (Kingma & Gao, 2024) and (Karras et al., 2022a). Let $p(\mathbf{x})$ be the data distribution and let $\lambda(t)$ define a *variance exploding* noise schedule with distribution $p(t)$ where $t \sim \mathcal{U}(0,1)$. Under this noise schedule we can define a noisy version of $\mathbf{x}$ at noise level $\sigma_t$ as

$$\mathbf{x}_t := \mathbf{x} + \sigma_t \boldsymbol{\epsilon} \quad \text{where} \quad \boldsymbol{\epsilon} \sim \mathcal{N}(0, \mathbf{I}). \tag{19}$$

Given noisy samples of data, the diffusion objective can be reduced to a weighted denoising objective,

$$\mathcal{L}_{\text{DPM}}(\boldsymbol{\epsilon}_\theta) = \frac{1}{2} \mathsf{E}_{p(t)p(\mathbf{x})q(\boldsymbol{\epsilon})}\left[w(t)\|\boldsymbol{\epsilon} - \boldsymbol{\epsilon}_\theta(\mathbf{x}_t; t)\|^2\right], \tag{20}$$

where $w(t)$ is a positive scalar-valued weighting function. Note that for the forward process defined in Eq. (19), the conditional score is $\mathbf{s}(\mathbf{x}_t|\mathbf{x}) = -\boldsymbol{\epsilon}/\sigma_t$. Thus, Eq. (20) can be interpreted as a weighted denoising score matching loss (Vincent, 2011) over multiple noise levels,

$$\mathcal{L}_{\text{DPM}}(\boldsymbol{\epsilon}_\theta) = \frac{1}{2} \mathsf{E}_{p(t)p(\mathbf{x})p(\mathbf{x}_t|\mathbf{x})}\left[w'(t)\left\|\mathbf{s}(\mathbf{x}_t|\mathbf{x}) + \frac{\boldsymbol{\epsilon}_\theta(\mathbf{x}_t; t)}{\sigma_t}\right\|^2\right], \tag{21}$$

where $w'(t) := \sigma_t^2 w(t)$ and the marginal score estimator is $\mathbf{s}_\theta(\mathbf{x}_t; t) := -\boldsymbol{\epsilon}_\theta(\mathbf{x}_t; t)/\sigma_t$.

**Sampling.** It is often beneficial to view DPMs as SDEs (Song et al., 2021b) where the forward process can be expressed as

$$\mathrm{d}\mathbf{x}_t = \mathbf{f}(\mathbf{x}_t, t)\mathrm{d}t + g(t)\mathrm{d}\mathbf{w},$$

where $\mathbf{w}$ is a standard Wiener process and $\mathbf{x}_0 = \mathbf{x}$. The time reversal of this process (i.e., the generative process) is known to follow the reverse SDE,

$$\mathrm{d}\mathbf{x}_t = \left[\mathbf{f}(\mathbf{x}_t, t) - g^2(t)\nabla_{\mathbf{x}_t}\log p(\mathbf{x}_t)\right]\mathrm{d}t + g(t)\mathrm{d}\bar{\mathbf{w}}.$$

Note that in practice $\nabla_{\mathbf{x}_t}\log p(\mathbf{x}_t)$ would be estimated by the score function $\mathbf{s}_\theta(\mathbf{x}_t; t)$ from a variant of DSM as in Eq. (21).

Sampling can be simulated through techniques such as annealed Langevin dynamics or ancestral sampling (Song et al., 2021b). While the above reverse SDE is stochastic in nature, there also exists a deterministic process known as the *probability flow ODE* that satisfies the same intermediate marginal distributions,

$$\mathrm{d}\mathbf{x}_t = \left[\mathbf{f}(\mathbf{x}_t, t) - \frac{1}{2}g^2(t)\nabla_{\mathbf{x}_t}\log p(\mathbf{x}_t)\right]\mathrm{d}t. \tag{22}$$

The benefit of the ODE formulation is that it can discretized more coarsely and hence sampling can done in fewer timesteps. Furthermore, sampling is possible by plugging in the updates from Eq. (22) into black-box ODE solvers, e.g., the Heun 2$^{\text{nd}}$ order solver (Karras et al., 2022a). Sampling can be sped even further if Eq. (22) can be solved exactly. Lu et al. (2022a) show that the exact solution to Eq. (22) at timestep $t$ given an initial value at timestep $s < t$ is,

$$\mathbf{x}_t = \mathbf{x}_s + 2\int_{\sigma_s}^{\sigma_t}\sigma_u\boldsymbol{\epsilon}_\theta\left(\mathbf{x}_u; u\right)\mathrm{d}\sigma_u. \tag{23}$$

Various samplers can be derived by approximating the exponentially weighted integral in different ways. For example, the widely used DDIM sampler (Song et al., 2021a) is an example of a first-order Taylor expansion of the integral term. At the core of all these algorithms is a score estimator/denoiser, which if learned accurately could improve the quality of samples produced.

**EDM Diffusion Architecture.** The EDM preconditioning diffusion model utilizes a base DDPM++ architecture from (Song et al., 2021b) for CIFAR-10 and the ADM architecture (Nichol & Dhariwal, 2021) for ImageNet $64 \times 64$. The EDM model uses a noise schedule that is defined as

$$\log\sigma_t \sim \mathcal{N}(-1.2, 1.2^2). \tag{24}$$

Rather than regressing against the unscaled additive noise as in DSM, EDM regresses against the original sample expressed in the following form,

$$\mathbf{x} = \frac{\sigma_{\text{data}}^2}{\sigma_t^2 + \sigma_{\text{data}}^2}\mathbf{x}_t + \frac{\sigma_t \cdot \sigma_{\text{data}}}{\sqrt{\sigma_t^2 + \sigma_{\text{data}}^2}}\mathbf{g}, \tag{25}$$

where $\sigma_{\text{data}} = 0.5$. To this end, EDM is parametrized with a denoising neural network,

$$\mathbf{f}_\theta(\mathbf{x}_t; t) = \frac{\sigma_{\text{data}}^2}{\sigma_t^2 + \sigma_{\text{data}}^2}\mathbf{x}_t + \frac{\sigma_t \cdot \sigma_{\text{data}}}{\sqrt{\sigma_t^2 + \sigma_{\text{data}}^2}}\mathbf{g}_\theta(\mathbf{x}_t; t), \tag{26}$$

which is trained by minimizing

$$\min_\theta \mathsf{E}_{p(\mathbf{x})q(\boldsymbol{\epsilon})p(t)}[\tilde{w}(t)\|\mathbf{x} - \mathbf{f}_\theta(\mathbf{x}_t; t)\|^2],$$

where

$$w_{\text{EDM}}(t) = \frac{\sigma_t\sigma_{\text{data}}}{\sqrt{\sigma_t^2 + \sigma_{\text{data}}^2}}. \tag{27}$$

This is equivalent to estimating $\mathbf{g}$ by minimizing the objective,

$$\mathcal{L}_{\text{EDM}}(\mathbf{g}_\theta) := \mathsf{E}_{p(\mathbf{x})q(\boldsymbol{\epsilon})p(t)}\left[\|\mathbf{g} - \mathbf{g}_\theta\left(\mathbf{x}_t; t\right)\|^2\right]. \tag{28}$$

Using Eq. (24) and Eq. (25) we can show that,

$$\mathbf{g} = \frac{\sqrt{\sigma_t^2 + \sigma_{\text{data}}^2}}{\sigma_t\sigma_{\text{data}}}\mathbf{x} - \frac{\sigma_{\text{data}}}{\sigma_t\sqrt{\sigma_t^2 + \sigma_{\text{data}}^2}}\mathbf{x}_t \tag{29}$$

$$= -\frac{\sqrt{\sigma_t^2 + \sigma_{\text{data}}^2}}{\sigma_{\text{data}}}\boldsymbol{\epsilon} + \frac{\sigma_t}{\sqrt{\sigma_t^2 + \sigma_{\text{data}}^2}\sigma_{\text{data}}}\mathbf{x}_t. \tag{30}$$

Therefore, in terms of Eq. (20) the EDM objective boils down to the unified diffusion objective with weighting function,

$$w(t) = \frac{\sigma_t^2 + \sigma_{\text{data}}^2}{\sigma_{\text{data}}^2}. \tag{31}$$

## B.2. Diffusion Distillation

Achieving state-of-the-art generation results on CIFAR-10 and ImageNet $64 \times 64$ using a Heun $2^{\text{nd}}$ order sampler with the EDM architecture requires 35 and 512 function evaluations (FEs) respectively. The goal of diffusion distillation is to distill a teacher model into a student model that can achieve high quality signal generation with few FEs.

The earliest works on distillation such as progressive distillation (Salimans & Ho, 2022) and knowledge distillation (Huang et al., 2023) train a student diffusion model with drastically reduced sampling budget to match the performance of a teacher model that is simulated in reverse. For example, given a teacher diffusion model parametrized as a denoiser $\mathbf{f}_\phi$ and a noisy sample $\mathbf{x}_t$, a "clean" target $\mathbf{x}_\phi^{(k)}$ is constructed by running the teacher model for $k$ steps in reverse. The student denoiser $\mathbf{f}_\theta$ is then optimized by minimizing the loss,

$$\mathcal{L}(\phi) := \mathsf{E}_{p(\mathbf{x})p(\mathbf{z})p(t)}[w(t)\|\mathbf{f}_\theta(\mathbf{x}_t; t) - \mathbf{x}_\phi^{(k)}\|^2].$$

Knowledge distillation on the other hand conditions the student model on intermediate features from the teacher diffusion model so as to regularize the learned weights more effectively and retain knowledge from the teacher model. These methods are expensive as it requires either simulating multiple steps of a teacher diffusion model or additionally probing it for feature extraction.

More recently a class of new diffusion distillation techniques grounded in reverse KL divergence minimization have gained popularity as discussed in Sec. 2. Diff-Instruct (Luo et al., 2024a), DMD (Yin et al., 2024b) and DMD2 (Yin et al., 2024a) all train a one-step generator $\mathbf{g}_\theta$ mapping noise $\mathbf{z} \sim \mathcal{N}(0, \mathbf{I})$ to generated samples by updating the generator in the direction of minimizing the reverse KLD,

$$\nabla_\theta \mathrm{D}_{\mathrm{KL}}^{\mathrm{avg}}(q_\theta\|p) = \mathsf{E}_{q(\mathbf{z})p(t)q(\boldsymbol{\epsilon})}[\nabla_\theta \mathbf{g}_\theta(\mathbf{z})(\mathbf{s}_{q_\theta}(\mathbf{x}_t) - \mathbf{s}_p(\mathbf{x}_t)) \mid_{\mathbf{x}=\mathbf{g}_\theta(\mathbf{z})}],$$

where $\mathbf{s}_p(\mathbf{x}_t) = \nabla_{\mathbf{x}_t} \log p(\mathbf{x}_t)$ and $\mathbf{s}_{q_\theta}(\mathbf{x}_t) = \nabla_{\mathbf{x}_t} \log q_\theta(\mathbf{x}_t)$. Assuming that the score model was learned using a parametrization similar to EDM, DMD scales the gradient and uses Tweedie's formula (Robbins, 1956) to express it in terms of a pretrained denoiser $\mathbf{f}_\phi$ and a denoiser for the fake samples $\mathbf{f}_\psi$,

$$\nabla_\theta \mathcal{L}_{\mathrm{DMD}}(\theta) = \mathsf{E}_{q(\mathbf{z})p(t)q(\boldsymbol{\epsilon})}[w_{\mathrm{DMD}}(\mathbf{x}_t, \mathbf{x}, t)\nabla_\theta \mathbf{g}_\theta(\mathbf{z})(\mathbf{f}_\psi(\mathbf{x}_t; t) - \mathbf{f}_\phi(\mathbf{x}_t; t)) \mid_{\mathbf{x}=\mathbf{g}_\theta(\mathbf{z})}],$$

where an adaptive weight is used to ensure that the scale of the gradient is roughly uniform across noise levels,

$$w_{\mathrm{DMD}}(\mathbf{x}_t, \mathbf{x}, t) := \frac{\sigma_t^2}{\|\mathbf{x} - \mathbf{f}_\phi(\mathbf{x}_t; t)\|_1}. \tag{32}$$

To mitigate mode collapse and enhance sample diversity, DMD employs an ODE-based regularizer by simulating the pretrained diffusion model in reverse. This process generates noise-image pairs, which are then used to further supervise the generator's training. However, collecting this dataset becomes prohibitively expensive for high-dimensional samples. To address this limitation, DMD2 introduces a GAN-based regularizer, which effectively minimizes the Jensen-Shannon divergence alongside the reverse KLD, or a variant of the forward KLD when implemented in a non-saturating manner. For further details on GAN training, refer to Appendix D.

Several methods build upon the divergence minimization framework by introducing regularizers based on alternative statistical distance measures. For instance, Moment Matching Distillation (MMD) (Salimans et al., 2024), Score Identity Distillation (SiD) (Zhou et al., 2024), and Score Implicit Matching (SiM) (Luo et al., 2024b) align the fake score model with the pretrained score model using a variant of the Fisher divergence:

$$\mathcal{L}_{\mathrm{Fisher}}(\psi) := \mathsf{E}_{q_\theta(\mathbf{x})p(t)q(\boldsymbol{\epsilon})}[w'(t)\|\mathbf{f}_\psi(\mathbf{x}_t; t) - \mathsf{sg}[\mathbf{f}_\phi(\mathbf{x}_t; t)]\|^2].$$

Here sg stands for the stop gradient operator. Additionally, both SiD and SiM extend this approach to generator training by minimizing the Fisher divergence, which requires a computationally expensive gradient calculation through the entire score model. To address this, they employ statistical approximations to make these gradient computations more practical.

## B.3. Consistency Models

Consistency models are a new class of generative models introduced by Song et al. (2023) that learn a consistency function between all points along the trajectory of the probability flow ODE of a reverse diffusion sampler. Concisely, given points

along one such trajectory, $\mathbf{x}_t, t \in [\epsilon, 1]$, where $\mathbf{x}_1 \sim \mathcal{N}(0, \mathbf{I})$, the consistency function satisfies,

$$\mathbf{f}(\mathbf{x}_t, t) = \begin{cases} \mathbf{x} & \text{if } t = \epsilon \\ \mathbf{f}(\mathbf{x}_s, s) & s \in [\epsilon, 1] \end{cases}$$

Given the boundary condition at the origin, the consistency function can be parametrized using a neural network similar to EDM ,

$$\mathbf{f}_\theta(\mathbf{x}_t, t) = \frac{\sigma_{\text{data}}^2}{(\sigma_t - \sigma_\epsilon)^2 + \sigma_{\text{data}}^2} \mathbf{x}_t + \frac{(\sigma_t - \sigma_\epsilon) \cdot \sigma_{\text{data}}}{\sqrt{\sigma_t^2 + \sigma_{\text{data}}^2}} \mathbf{g}_\theta(\mathbf{x}_t; t).$$

Given a noisy sample $\mathbf{x}_t = \mathbf{x} + \sigma_t \epsilon, \epsilon \sim \mathcal{N}(0, \mathbf{I})$, first a single step of the probability flow ODE is simulated using the Euler sampler by running one step of sampling using Eq. (23),

$$\mathbf{x}_s = \mathbf{x}_t + (t - s)t \nabla_{\mathbf{x}_t} \log p(\mathbf{x}_t)$$

This can be computed using either a pretrained score model or via a single sample Monte-Carlo estimate. In the latter setting, it is important that the timesteps $s$ and $t$ are very close to each other for the approximation to hold. In consistency distillation a pretrained score model $\mathbf{s}_\phi$ is available and a single sampling step along the PF-ODE is simulated as

$$\mathbf{x}_s^\phi = \mathbf{x}_t + (t - s)t\mathbf{s}_\phi(\mathbf{x}_t; t).$$

Then the consistency function is learned by minimizing

$$\mathcal{L}_{\text{CD}}(\theta) = \mathsf{E}_{p(\mathbf{x})q(\epsilon)p(t)}[w(t)d(\mathbf{f}_\theta(\mathbf{x}_t; t), \text{sg}[\mathbf{f}_\theta(\mathbf{x}_{t-\Delta t}^\phi; t - \Delta t)])],$$

where $d$ is some distance measure, $w(t)$ is some positive weighting function and $s = t - \Delta t$, with $\Delta t$ some fixed timestep difference. Song et al. (2023) initially proposed using the LPIPS distance but subsequent works (Song & Dhariwal, 2024a; Geng et al., 2025) have shown that similar performance can be achieved by using the $\ell_2$ distance or a pseudo-Huber norm.

Unlike distillation techniques, consistency models can also be trained from scratch. Assume that $s = t - \delta t, \delta t \to 0$. Then, the sampling step can be approximated using Tweedie's formula (Robbins, 1956),

$$\mathbf{x}_s \approx \mathbf{x}_t + (t - s)\frac{\mathbf{x} - \mathbf{x}_t}{t}$$
$$= \mathbf{x} + s\epsilon.$$

Thus, the consistency function can now be learned by minimizing,

$$\mathcal{L}_{\text{CT}}(\theta) = \mathsf{E}_{p(\mathbf{x})q(\epsilon)p(t)}[w(t)d(\mathbf{f}_\theta(\mathbf{x} + t\epsilon; t), \text{sg}[\mathbf{f}_\theta(\mathbf{x} + (t - \delta t)\epsilon; t - \delta t)])],$$

Consistency distillation still lags behind distillation methods based on reverse KL minimization, but consistency training often demonstrates more impressive results. However, consistency training is still inherently unstable and requires careful design of both the noise schedule due to limiting nature of $\delta t$ and distance measure (Song & Dhariwal, 2024a; Geng et al., 2025). Stabilizing and making this objective simpler is the focus of a lot of current research in the area.

## C. Detailed Description of Score-of-Mixture Training and Distillation

### C.1. Amortized Denoiser

Modern diffusion architectures such as the EDM architecture (Karras et al., 2022b) are specially designed for denoising purposes (see Appendix B). Hence, in practice we choose to train an *amortized denoiser*, $\mathbf{f}_\psi(\mathbf{x}_t; \alpha, t) \approx \mathsf{E}_{\alpha p_t + (1-\alpha)q_{\theta,t}}[\mathbf{x}|\mathbf{x}_t]$, upon which the amortized score can be recovered using Tweedie's formula (Robbins, 1956),

$$\mathbf{s}_\psi(\mathbf{x}_t; \alpha, t) = -\frac{1}{\sigma_t^2}\mathbf{x}_t + \frac{1}{\sigma_t^2}\mathbf{f}_\psi(\mathbf{x}_t; \alpha, t).$$

The mixture score matching loss in Eq. (8) can be expressed with this denoiser as

$$\mathcal{L}_{\text{gen}}^{\text{denoise}}(\psi; \alpha, t) := \alpha\,\mathsf{E}_{p(\mathbf{x})q(\epsilon)}[\|\mathbf{f}_\psi(\mathbf{x}; \alpha, t) - \mathbf{x}\|^2] + (1 - \alpha)\,\mathsf{E}_{q_\theta(\mathbf{x})q(\epsilon)}[\|\mathbf{f}_\psi(\mathbf{x}; \alpha, t) - \mathbf{x}\|^2].$$

### C.2. Score-of-Mixture Training

Here, we present a pseudocode for Score-of-Mixture Training (SMT). See Algorithm 1.

---

**Algorithm 1** Score-of-Mixture Training

---

**Inputs:** Randomly initialized generator $\mathbf{g}_\theta$, amortized score model $\mathbf{s}_\psi$, discriminator $\ell_\psi$, real dataset $\mathcal{D}$, score training sub-iterations = 5, learning rates $(\eta_{\text{gen}}, \eta_{\text{score}})$, GAN regularizer weights (score = $\mu$, gen = $\lambda$)

**Pretraining:** Train $\mathbf{g}_\theta$ with DSM using $\mathcal{D}$
**for** each pretraining iteration **do**
    Sample mini-batch $\mathbf{x} \sim \mathcal{D}$ and add noise $\mathbf{x}_t = \mathbf{x} + \sigma_t \boldsymbol{\epsilon}, \boldsymbol{\epsilon} \sim \mathcal{N}(0, \mathbf{I})$
    Compute DSM loss $\mathcal{L}_{\text{DSM}}(\theta)$ (see Sec. 2)
    Update parameters: $\theta \leftarrow \theta - \eta_{\text{DSM}} \nabla_\theta \mathcal{L}_{\text{DSM}}(\theta)$
**end for**

**Training:** Alternating updates of $\mathbf{g}_\theta$ and $\mathbf{s}_\psi$
**for** each training iteration **do**

    **Generator Training:** Freeze $\mathbf{s}_\psi$
    Sample mini-batch of fake samples $\mathbf{x}^{\text{fake}} = \mathbf{g}_\theta(\mathbf{z}), \mathbf{z} \sim \mathcal{N}(0, \mathbf{I})$
    Sample $t \sim p(t)$ and $\alpha$ as described in Sec. 3.5
    Compute weighted generator gradient $\gamma_\psi^w(\theta; \alpha, t)$ from Eq. (9)
    Compute GAN regularizer loss $\mathcal{L}_{\text{GAN}}^{(\alpha,t)}$ from Eq. (11)
    Update parameters:

$$\theta \leftarrow \theta - \eta_{\text{gen}} \, \mathsf{E}_{p(\alpha)p(t)}[w(\mathbf{x}_t^{\text{fake}}, \mathbf{x}^{\text{fake}}, \alpha, t)\gamma_\psi^w(\theta; \alpha, t) + \lambda \nabla_\theta \mathcal{L}_{\text{GAN}}^{(\alpha,t)}(\theta)]$$

    **Amortized Score Training:** Freeze $\mathbf{g}_\theta$
    **for** each sub-iteration **do**
        Sample mini-batch of real samples $\mathbf{x}^{\text{real}} \sim \mathcal{D}$
        Sample $t \sim p(t)$ and $\alpha$
        Compute score matching loss $\mathcal{L}_{\text{score}}(\psi; \alpha, t)$ from Eq. (8)
        Compute non-saturated discriminator loss $\mathcal{L}_{\text{disc}}(\psi)$ (see Appendix D)
        Update parameters:

$$\psi \leftarrow \psi - \eta_{\text{score}} \nabla_\psi (\mathsf{E}_{p(\alpha)p(t)}[\mathcal{L}_{\text{score}}(\psi; \alpha, t)] + \mu \mathcal{L}_{\text{disc}}(\psi))$$

    **end for**
**end for**

**Return:** Trained model parameters $\theta, \psi$

---

### C.3. Score-of-Mixture Distillation

We present an overview figure and pseudocode of Score-of-Mixture Distillation in Fig. 4 and Alg. 2. We highlight the central differences in the distillation training from the training from scratch in Alg. 1.

- The pretrained score model on the real data is available during distillation. Thus, rather than defining an amortized score model that takes in a conditioning variable for $\alpha$, we show that we only need to learn a score model on the fake samples to make the objective simpler (see Proposition A.2). This is more effective as learning the score of multiple interpolated distributions is a generally more challenging task.

- We can initialize the fake score and generator with the weights from the pretrained model and don't need to run a separate pretraining phase as in training from scratch.

- Notice that in amortized score training in the distillation setting there is no explicit GAN loss to learn the discriminator in Algorithm 2. Learning the discriminator implicit to our proposed parametrization and once learned we can use this to minimize the skewed divergence via a non-saturating GAN regularizer for generator training.

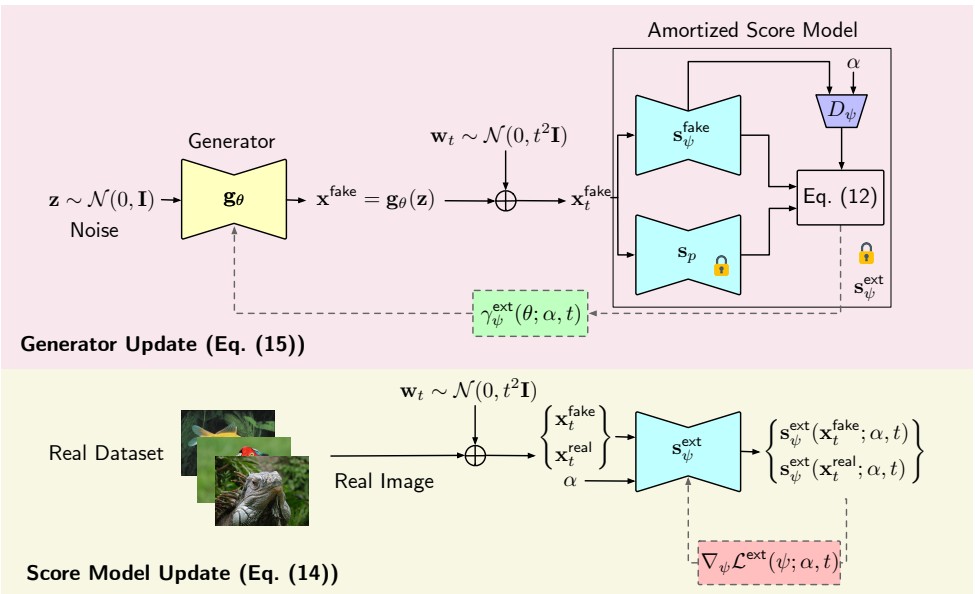

*Figure 4.* Overview of Score-of-Mixture Distillation. **Top:** To update the generator weights, the fake image is diffused at noise level $t$ and then used to compute the gradient of the $\alpha$-skew divergence with the explicitly parametrized amortized score model using Eq. (15). **Bottom:** Amortized score model training involves computing the score of the mixture distribution on both fake and real samples diffused with noise level $t$ and then updating the weights using the gradient of Eq. (14).

---

**Algorithm 2** Score-of-Mixture Distillation

---

**Inputs:** Randomly initialized generator $\mathbf{g}_\theta$, fake score model $\mathbf{s}_\psi^{\text{fake}}$, discriminator $\ell_\psi$, pretrained score model $\mathbf{s}_p$, real dataset $\mathcal{D}$, score training sub-iterations = 5, learning rates $(\eta_{\text{gen}}, \eta_{\text{score}})$, GAN generator regularizer weight $\lambda$

**Initialization** Initialize $\mathbf{s}_\psi^{\text{fake}}$ and $\mathbf{g}_\theta$ with weights from $\mathbf{s}_p$

**Training:** Alternating updates of $\mathbf{g}_\theta$ and $\mathbf{s}_\psi$
**for** each training iteration **do**

    **Generator Training:** Freeze $\mathbf{s}_\psi^{\text{fake}}$ and $\ell_\psi$
    Sample mini-batch of fake samples $\mathbf{x}^{\text{fake}} = \mathbf{g}_\theta(\mathbf{z}), \mathbf{z} \sim \mathcal{N}(0, \mathbf{I})$
    Sample $t \sim p(t)$ and $\alpha$ as described in Sec. 3.5
    Compute generator gradient $\gamma_\psi^{\text{ext}}(\theta; \alpha, t)$ from Eq. (15)
    Compute GAN regularizer loss $\mathcal{L}_{\text{GAN}}^{(\alpha,t)}$ from Eq. (11)
    Update parameters:
$$\theta \leftarrow \theta - \eta_{\text{gen}} \, \mathsf{E}_{p(\alpha)p(t)}[\gamma_\psi^{\text{ext}}(\theta; \alpha, t) + \lambda \nabla_\theta \mathcal{L}_{\text{GAN}}^{(\alpha,t)}(\theta)]$$

    **Amortized Score Training:** Freeze $\mathbf{g}_\theta$
    **for** each sub-iteration **do**
        Sample mini-batch of real samples $\mathbf{x}^{\text{real}} \sim \mathcal{D}$
        Sample $t \sim p(t)$ and $\alpha$
        Compute score matching loss using explicit parametrization $\mathcal{L}_{\text{score}}^{\text{ext}}(\psi; \alpha, t)$ from Eq. (14)
        Update parameters:
$$\psi \leftarrow \psi - \eta_{\text{score}} \nabla_\psi \, \mathsf{E}_{p(\alpha)p(t)}[\mathcal{L}_{\text{score}}^{\text{ext}}(\psi; \alpha, t)]$$

    **end for**
**end for**

**Return:** Trained model parameters $\theta, \psi$

---

## D. On GAN Training

The vanilla GAN is

$$\min_{\theta} \max_{\psi} \{ \mathsf{E}_{p(\mathbf{x})}[\log D_{\psi}(\mathbf{x})] + \mathsf{E}_{q_{\theta}(\mathbf{x})}[\log(1 - D_{\psi}(\mathbf{x}))] \}.$$

Breaking down, the discriminator training is

$$\min_{\psi} - \mathsf{E}_{p(\mathbf{x})}[\log D_{\psi}(\mathbf{x})] - \mathsf{E}_{q_{\theta}(\mathbf{x})}[\log(1 - D_{\psi}(\mathbf{x}))] \},$$

and the generator training is

$$\min_{\theta} \mathsf{E}_{q_{\theta}(\mathbf{x})}[\log(1 - D_{\psi}(\mathbf{x}))].$$

Note that

$$D_{\psi}(\mathbf{x}) = \frac{r_{\psi}(\mathbf{x})}{1 + r_{\psi}(\mathbf{x})} = \frac{1}{1 + r_{\psi}^{-1}(\mathbf{x})}.$$

Further, the sigmoid logit $C_{\psi}(\mathbf{x})$, i.e., $D_{\psi}(\mathbf{x}) = \sigma(C_{\psi}(\mathbf{x}))$, is $\log r_{\psi}(\mathbf{x})$. Note that the optimal discriminator for each $\theta$ is $D^{\star}(\mathbf{x}) = \frac{p(\mathbf{x})}{p(\mathbf{x}) + q_{\theta}(\mathbf{x})}$ or $r^{\star}(\mathbf{x}) = \frac{p(\mathbf{x})}{q_{\theta}(\mathbf{x})}$.

The non-saturating versions is training generator based on

$$\min_{\theta} - \mathsf{E}_{q_{\theta}(\mathbf{x})}[\log D_{\psi}(\mathbf{x})] \approx \mathsf{E}_{q_{\theta}(\mathbf{x})}\left[\log\left(\frac{q_{\theta}(\mathbf{x})}{p(\mathbf{x})} + 1\right)\right].$$

The StyleGAN uses the non-saturating loss:

$$\min_{\psi} \ \mathsf{E}_{p(\mathbf{x})}[\mathsf{sp}(-\log r_{\psi}(\mathbf{x}))] + \mathsf{E}_{q_{\theta}(\mathbf{x})}[\mathsf{sp}(\log r_{\psi}(\mathbf{x}))] \},$$

$$\min_{\theta} \ \mathsf{E}_{q_{\theta}(\mathbf{x})}[\mathsf{sp}(-\log r_{\psi}(\mathbf{x}))].$$

Note that $\mathsf{sp}(y) := \log(1 + e^{y})$. Hence, note that

$$\mathsf{sp}(-\log r_{\psi}(\mathbf{x})) = \log(1 + r_{\psi}^{-1}(\mathbf{x})) = -\log D_{\psi}(\mathbf{x}),$$

$$\mathsf{sp}(\log r_{\psi}(\mathbf{x})) = \log(1 + r_{\psi}(\mathbf{x})) = -\log(1 - D_{\psi}(\mathbf{x})).$$

### D.1. On the Non-Saturating Generative Loss

The original, saturating version of the generator objective is

$$\min_{\theta} \mathsf{E}_{q_{\theta}(\mathbf{x})}[-\mathsf{sp}(\log r_{\psi}(\mathbf{x}))] = \mathsf{E}_{q_{\theta}(\mathbf{x})}[-\log(1 + r_{\psi}(\mathbf{x}))],$$

whose gradient is

$$\nabla_{\theta} \mathsf{E}_{q(\mathbf{z})}[-\log(1 + r_{\psi}(\mathbf{g}_{\theta}(\mathbf{z})))] = \mathsf{E}_{q(\mathbf{z})}\left[-\frac{r_{\psi}'(\mathbf{g}_{\theta}(\mathbf{z}))}{1 + r_{\psi}(\mathbf{g}_{\theta}(\mathbf{z}))}\nabla_{\theta}\mathbf{g}_{\theta}(\mathbf{z})\right].$$

Note that the *plug-in* reverse KL-divergence loss is

$$\min_{\theta} \mathsf{E}_{q_{\theta}(\mathbf{x})}[-\log r_{\psi}(\mathbf{x})].$$

Compared to this, the non-saturating loss has the additional $\mathsf{sp}(\cdot)$:

$$\min_{\theta} \mathsf{E}_{q_{\theta}(\mathbf{x})}[\mathsf{sp}(-\log r_{\psi}(\mathbf{x}))] = \mathsf{E}_{q_{\theta}(\mathbf{x})}[\log(1 + r_{\psi}(\mathbf{x})^{-1})].$$

This seems to help prevent vanishing gradients. Consider

$$\min_{\theta} \mathsf{E}_{q_{\theta}(\mathbf{x})}[\log(\tau + r_{\psi}(\mathbf{x})^{-1})] = \mathsf{E}_{q(\mathbf{z})}[\log(\tau + r_{\psi}(\mathbf{g}_{\theta}(\mathbf{z}))^{-1})],$$

If $\tau = 0$, it boils down to the plug-in reverse KL divergence, and $\tau = 1$ recovers the non-saturating loss. If we consider a gradient with respect to $\theta$, we get

$$\nabla_\theta \mathsf{E}_{q(\mathbf{z})}[\log(\tau + r_\psi(\mathbf{g}_\theta(\mathbf{z}))^{-1})] = \mathsf{E}_{q(\mathbf{z})}\left[-\frac{r'_\psi(\mathbf{g}_\theta(\mathbf{z}))}{r_\psi(\mathbf{g}_\theta(\mathbf{z}))(1 + \tau r_\psi(\mathbf{g}_\theta(\mathbf{z})))}\nabla_\theta \mathbf{g}_\theta(\mathbf{z})\right]$$

Here, recall that $r_\psi(\mathbf{x}) \approx \frac{p(\mathbf{x})}{q_\theta(\mathbf{x})}$ is supposed to be small for generated samples $\mathbf{x} = \mathbf{g}_\theta(\mathbf{z})$. Therefore, the plug-in loss with $\tau = 0$ is inherently prone to vanishing gradient.

### D.2. GAN-Type Regularization with $\alpha$-JSD

To train a discriminator for our GAN-type regularization, we opt to use a modified GAN discriminator objective defined as

$$\min_\psi -\alpha\, \mathsf{E}_{p(\mathbf{x})}[\log D_\psi(\mathbf{x}; \alpha)] - (1 - \alpha)\, \mathsf{E}_{q_\theta(\mathbf{x})}[\log(1 - D_\psi(\mathbf{x}; \alpha))].$$

Similar to the vanilla GAN, the optimal discriminator for each $\theta$ and $\alpha$ in this case is

$$D_\psi^\star(\mathbf{x}; \alpha) = \frac{\alpha p(\mathbf{x})}{\alpha p(\mathbf{x}) + (1 - \alpha)q_\theta(\mathbf{x})}$$

Then, the $\alpha$-JSD can be approximated as

$$\mathrm{D}_{\mathrm{JSD}}^{(\alpha)}(q_\theta, p) \approx \frac{1}{1 - \alpha}\, \mathsf{E}_{p(\mathbf{x})}\left[\log \frac{D_\psi(\mathbf{x}; \alpha)}{\alpha}\right] + \frac{1}{\alpha}\, \mathsf{E}_{q_\theta(\mathbf{x})}\left[\log \frac{1 - D_\psi(\mathbf{x}; \alpha)}{1 - \alpha}\right].$$

Hence, with this approximation, the generator update can be done via

$$\min_\theta \frac{1}{\alpha}\, \mathsf{E}_{q_\theta(\mathbf{x})}\left[\log \frac{1 - D_\psi(\mathbf{x}; \alpha)}{1 - \alpha}\right].$$

In practice, we can use a weighted non-saturating version of the loss as well,

$$\min_\theta -\, \mathsf{E}_{q_\theta(\mathbf{x})}\left[\log \frac{D_\psi(\mathbf{x}; \alpha)}{\alpha}\right] = \min_\theta \mathsf{E}_{q_\theta(\mathbf{x})}\left[\mathrm{sp}\left(-\ell_\psi(\mathbf{x}) - \log \frac{\alpha}{1 - \alpha}\right)\right] + \log \alpha,$$

where $\ell_\psi(\mathbf{x}) = \log \frac{p(\mathbf{x})}{q_\theta(\mathbf{x})}$.

### D.3. On Discriminator Training with Mixture Score Matching Loss

In Sec. 4, we plugged in the explicit parameterization

$$\mathbf{s}_\psi^{\mathrm{exp}}(\mathbf{x}; \alpha) := D_\psi(\mathbf{x}; \alpha)\mathbf{s}_p(\mathbf{x}) + (1 - D_\psi(\mathbf{x}; \alpha))\mathbf{s}_\psi^{\mathrm{fake}}(\mathbf{x}),$$

to the mixture regression loss in Eq. (7), to train the fake score and the discriminator simultaneously. If we consider an ideal scenario where we have the perfect score models for both $p$ and $q$, then all we need to train is the discriminator and the mixture regression objective can be interpreted as a discriminator objective. Here we reveal its connection to an instance of $f$-GAN discriminator objective.

Let $\mathbf{s}_p(\mathbf{x})$ and $\mathbf{s}_q(\mathbf{x})$ be the underlying score functions for $p$ and $q$, respectively. Then, the explicit parameterization becomes

$$\mathbf{s}_\psi^{\mathrm{exp}}(\mathbf{x}; \alpha) = D_\psi(\mathbf{x}; \alpha)\mathbf{s}_p(\mathbf{x}) + (1 - D_\psi(\mathbf{x}; \alpha))\mathbf{s}_q(\mathbf{x}),$$

and the mixture regression objective becomes only a function of the discriminator, i.e.,

$$\begin{aligned}
\mathcal{L}(\psi; \alpha) &= \alpha\, \mathsf{E}_{p(\mathbf{x})}[\|\mathbf{s}_\psi^{\mathrm{exp}}(\mathbf{x}; \alpha) - \mathbf{s}_p(\mathbf{x})\|^2] + (1 - \alpha)\, \mathsf{E}_{q_\theta(\mathbf{x})}[\|\mathbf{s}_\psi^{\mathrm{exp}}(\mathbf{x}; \alpha) - \mathbf{s}_q(\mathbf{x})\|^2] \\
&= \alpha\, \mathsf{E}_{p(\mathbf{x})}[(1 - D_\psi(\mathbf{x}; \alpha))^2\|\mathbf{s}_p(\mathbf{x}) - \mathbf{s}_q(\mathbf{x})\|^2] + (1 - \alpha)\, \mathsf{E}_{q(\mathbf{x})}[D_\psi(\mathbf{x}; \alpha)^2\|\mathbf{s}_p(\mathbf{x}) - \mathbf{s}_q(\mathbf{x})\|^2] \\
&= \int \left\{\alpha p(\mathbf{x})(1 - D_\psi(\mathbf{x}; \alpha))^2 + (1 - \alpha)q(\mathbf{x})D_\psi(\mathbf{x}; \alpha)^2\right\}\|\mathbf{s}_p(\mathbf{x}) - \mathbf{s}_q(\mathbf{x})\|^2\, d\mathbf{x}.
\end{aligned}$$

Here, we note that the term $\|\mathbf{s}_p(\mathbf{x}) - \mathbf{s}_q(\mathbf{x})\|^2$ is common in both expectation, and can be safely dropped to train the discriminator, which leads to a simplified objective

$$\mathcal{L}'(\psi; \alpha) = \alpha\, \mathsf{E}_{p(\mathbf{x})}[(1 - D_\psi(\mathbf{x}; \alpha))^2] + (1 - \alpha)\, \mathsf{E}_{q(\mathbf{x})}[D_\psi(\mathbf{x}; \alpha)^2]$$
$$= \int \left\{ \alpha p(\mathbf{x})(1 - D_\psi(\mathbf{x}; \alpha))^2 + (1 - \alpha)q(\mathbf{x})D_\psi(\mathbf{x}; \alpha)^2 \right\} d\mathbf{x}.$$

We note that this is equivalent to the discriminator objective induced by the following $f$-divergence

$$D_{f_\alpha}(p \parallel q) := 1 - \int \frac{p(\mathbf{x})q(\mathbf{x})}{\alpha p(\mathbf{x}) + (1 - \alpha)q(\mathbf{x})}\, d\mathbf{x} := D_{\alpha\text{-LC}}(p \parallel q),$$

where $f_\alpha(r) := \frac{(1-\alpha)(1-r)}{\alpha r + (1-\alpha)}$ is a convex function over $[0, \infty)$ for $\alpha \in (0, 1)$. For $\alpha = \frac{1}{2}$, this divergence becomes symmetric in $p$ and $q$ and is known as the Le Cam distance (Le Cam, 2012, p. 47) in the literature (Polyanskiy & Wu, 2019). We thus call the general divergence for $\alpha \in (0, 1)$ the $\alpha$-Le Cam distance. In the GAN literature, this is known as the LSGAN objective (Mao et al., 2017).

As we revealed, our discriminator training in distillation can also be done separately using the $\alpha$-Le Cam-distance-based objective. However, we conjecture that our score-regression-based end-to-end objective may have benefit, as our primary goal of discriminator training is to use it in the generator update in the form of an approximate score of mixture. We leave the further exploration of such alternative methods as a future work.

# E. More on Experiments and Additional Results

We present some additional experiments and results in this section. We first provide a more detailed training configuration for our experiments in Sec. 5 and then evaluate our proposed method on a synthetic swiss-roll dataset in Appendix E.2. Finally, we present some samples generated from Score-of-Mixture Training and Score-of-Mixture Distillation in Figs. 9-12.

## E.1. Training Configuration

We summarize the detailed training configuration in Table 3.

Table 3. Hyperparameters used for training one-step generators with Score-of-Mixture Training and Distillation.

| Hyperparameter | CIFAR-10 | | ImageNet $64 \times 64$ | |
|---|---|---|---|---|
| | Scratch | Distillation | Scratch | Distillation |
| Generator learning rate | 1e-4 | 5e-5 | 5e-6 | 2e-6 |
| Score learning rate | 5e-4 | 5e-5 | 5e-5 | 2e-6 |
| Score learning rate decay | cosine | None | cosine | None |
| Batch size | 280 | 280 | 280 | 280 |
| Diffusion pretraining steps | 15k | N/A | 40k | N/A |
| Training iterations | 150k | 150k | 200k | 200k |
| Score dropout probability | 0.13 | 0.00 | 0.00 | 0.00 |
| Number of GPUs | $2 \times$ A100 | $4\times$ A100 | $7\times$ A100 | $7\times$ A100 |

## E.2. Toy Swiss Roll

We tested our proposed framework and ablated various design choices on a synthetic swiss roll dataset. We followed the dataset setup by Che et al. (2020). We trained models with SMT and SMD and compared this against an amortized version of reverse KL minimization with DMD weighting ($\alpha \in \{0, 1\}$) similar to the ablations in Sec. 5.3. Additionally we compared against non-score-based baselines including the vanilla GAN and Diffusion-GAN (Wang et al., 2023).

Across all experiments, we use the same generator architecture — a two-layer MLP with a hidden dimension of 128 and leaky ReLU nonlinearity. We train all models for 200k steps on a single NVIDIA 3090 GPU with a batch size of 256. All score-based methods leverage a learning rate of 1e-5 for the generator and 1e-4 for the amortized score (and discriminator when applicable) whereas the GAN-based methods use a learning rate of 1e-4 for both generator and discriminator. We use the AdamW optimizer without any learning rate schedulers.

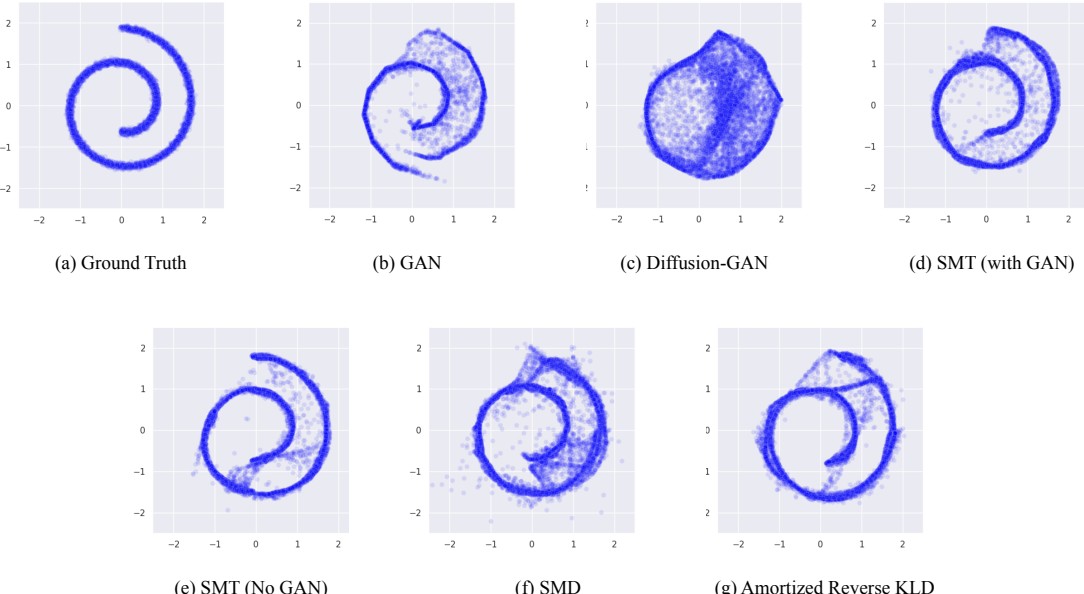

*Figure 5.* Samples produced by generators trained using different methods. All figures are created using 10,000 samples from the respective generator.

The samples produced are shown in Fig. 5. Notice how the GAN is unable to perfectly cover the entire continuous mode of the swiss roll. The Diffusion-GAN, a multi-noise level extension of the GAN, covers the mode but also samples from areas of low density. We found the latter to be sensitive to the chosen noise levels in comparison to the methods based on updating the generator using the score.

Our results for training from scratch and distillation are presented in Fig. 5d-f. All three methods successfully capture the modes of the underlying distribution. While the impact of the GAN regularizer is less pronounced than in our high-dimensional experiments, we observe that enabling it (as in Fig. 5d) reduces the number of samples in low-density regions compared to Fig. 5e. The distillation results in Fig. 5f appear slightly noisy, likely due to the quality of the pre-trained score model. This highlights the advantage of training from scratch, as it avoids amplifying existing estimation errors in the pre-trained model.

In Fig. 5g, we present an ablation result of the $\alpha$-sampler, by using only $\alpha = 1$. This corresponds to minimizing reverse KLD as in DMD and DMD2. Unlike in the high-dimensional setting presented in Fig. 2b, we observe that using the single $\alpha = 1$ produces visually plausible samples in this low-dimensional synthetic example. However, our method in Fig. 5d (i.e., SMT with GAN regularizer) produces fewer spurious samples compared to Fig. 5g, suggesting the benefit of multiple $\alpha$ values.

### E.3. Image Interpolation

The one-step generator is a mapping between the representation space, which in this case is the space of standard multivariate Gaussian variables, and the space of images. Thus, to study the representation space we followed the approached in (Song et al., 2023) and spherically interpolated between two randomly chosen noise instances $\mathbf{z}_0$ and $\mathbf{z}_1$,

$$\mathbf{z}_\beta = \frac{\sin((1-\beta)\psi)}{\sin(\psi)}\mathbf{z}_0 + \frac{\sin(\beta\psi)}{\sin(\psi)}\mathbf{z}_1,$$

where $\beta \in [0, 1]$ and $\psi = \arccos\left(\frac{\mathbf{z}_0^\mathsf{T}\mathbf{z}_1}{\|\mathbf{z}_0\|_2\|\mathbf{z}_1\|_2}\right)$. After interpolating between these two points, the interpolated image can be obtained as $\mathbf{x}_\beta = \mathbf{g}_\theta(\mathbf{z}_\beta)$ as shown in Figures 6 and 7.

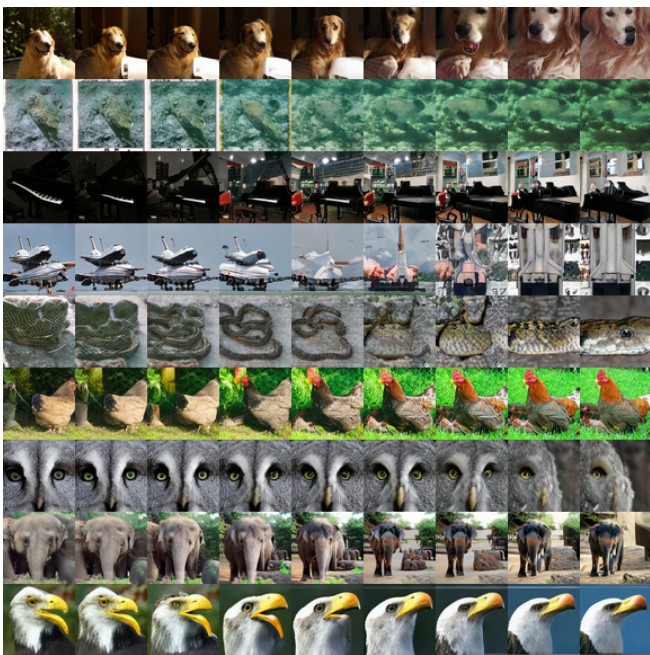

*Figure 6.* Visualizing the latent space of the one-step generator trained on ImageNet $64 \times 64$ by interpolating between two noise inputs. The leftmost and rightmost image in each row correspond to synthesized images with the same class and different noises $\mathbf{z}_0$ and $\mathbf{z}_1$. All intermediate images are obtained by applying the generator on a spherical interpolation between these noise instances, demonstrating the interpretable learned latent space of the generator.

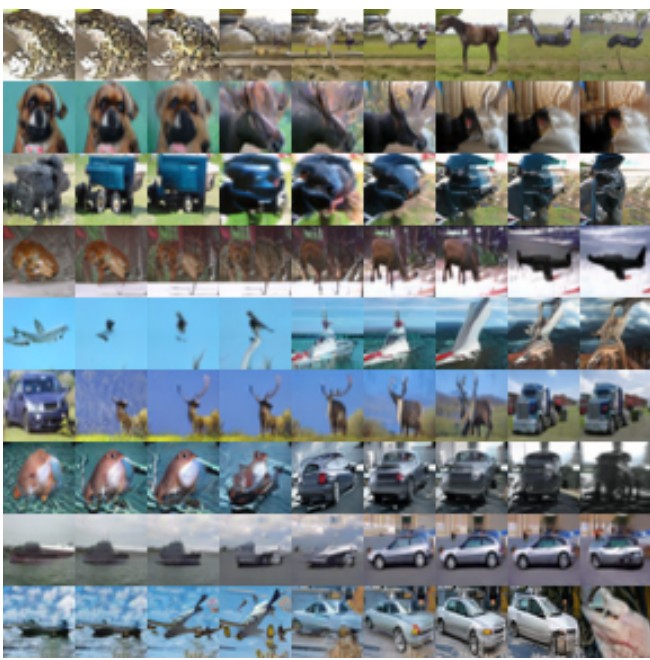

*Figure 7.* Visualizing the latent space of the one-step generator trained on the CIFAR-10 dataset by interpolating between two noise inputs. The leftmost and rightmost image in each row correspond to different noises $\mathbf{z}_0$ and $\mathbf{z}_1$. All intermediate images are obtained by applying the generator on a spherical interpolation between these noise instances, demonstrating the interpretable learned latent space of the generator.

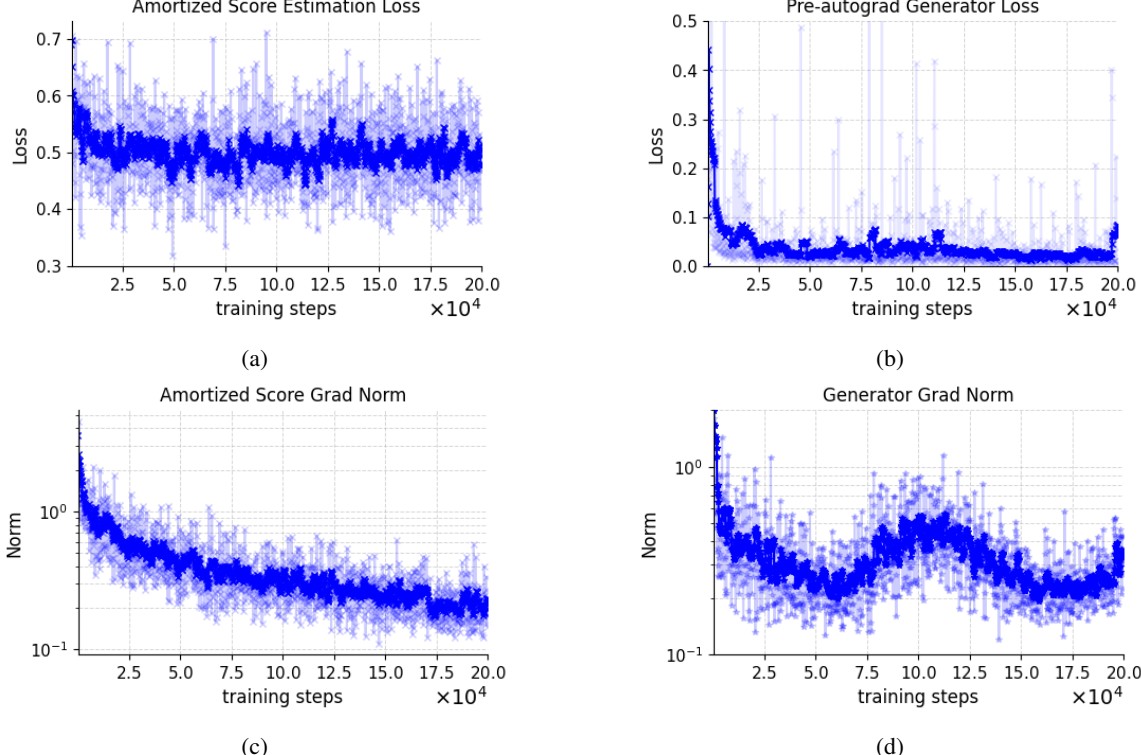

*Figure 8.* SMT training curves on ImageNet 64x64. Plotted in dark blue is the running average trajectory of the different metrics. Both the loss curves in (a) and (b) are smooth and do not explode. This is further supported by the curves of the respective gradient norms in (c) and (d), where it is clear the underlying optimization is stable and no gradient explosion occurs. The actual generator loss involved the gradient of the generator multiplied by the difference of scores. During implementation we let autograd take care of this gradient and the loss that is calculated pre-autograd is shown in (b).

### E.4. Additional Training Curves

To further demonstrate the stable training dynamics of SMT, we provide additional training curves in Figure 8 showing a consistent decrease in both loss and gradient norm. Notably, at no point do we observe any spikes or instability in either metric. Since we compute the generator's gradient directly, the plotted loss in Figure 8b corresponds to the proxy objective prior to applying automatic differentiation.

### E.5. Samples

We present some image samples generated by SMT and SMD in Figs. 9-12.

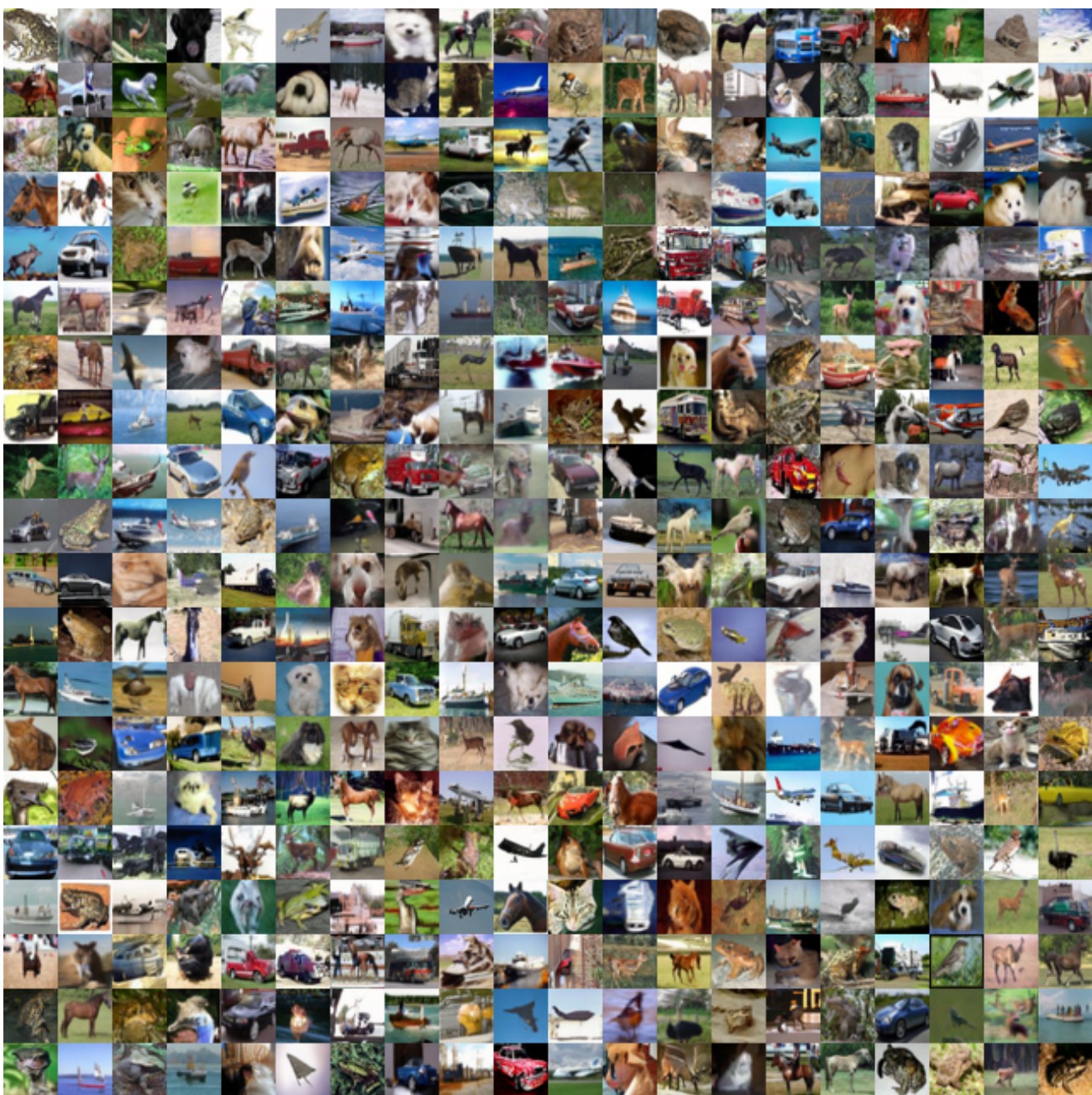

*Figure 9.* One-step generated samples from SMT on CIFAR-10 (unconditional).

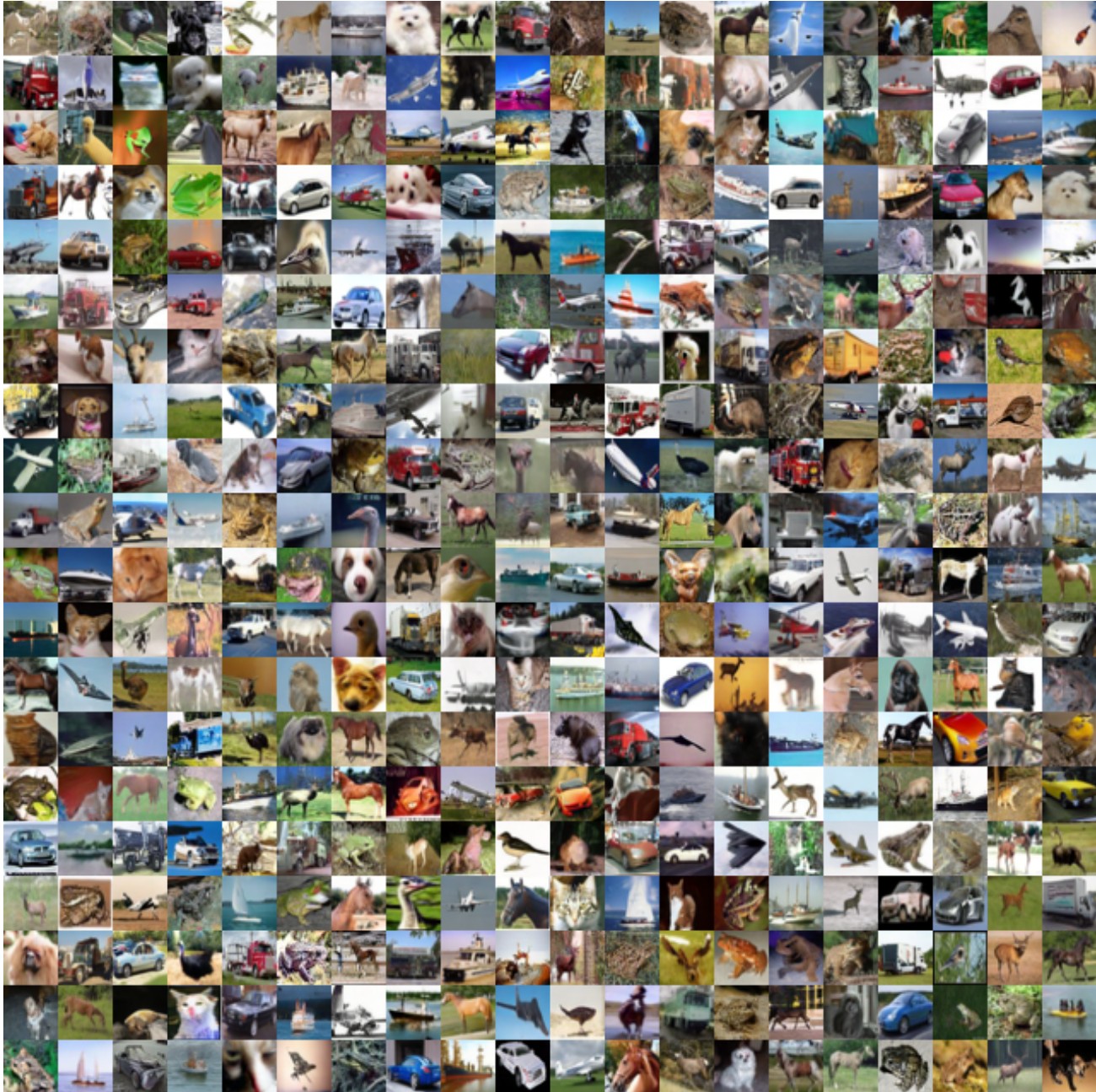

*Figure 10.* One-step generated samples from SMD on CIFAR-10 (unconditional).

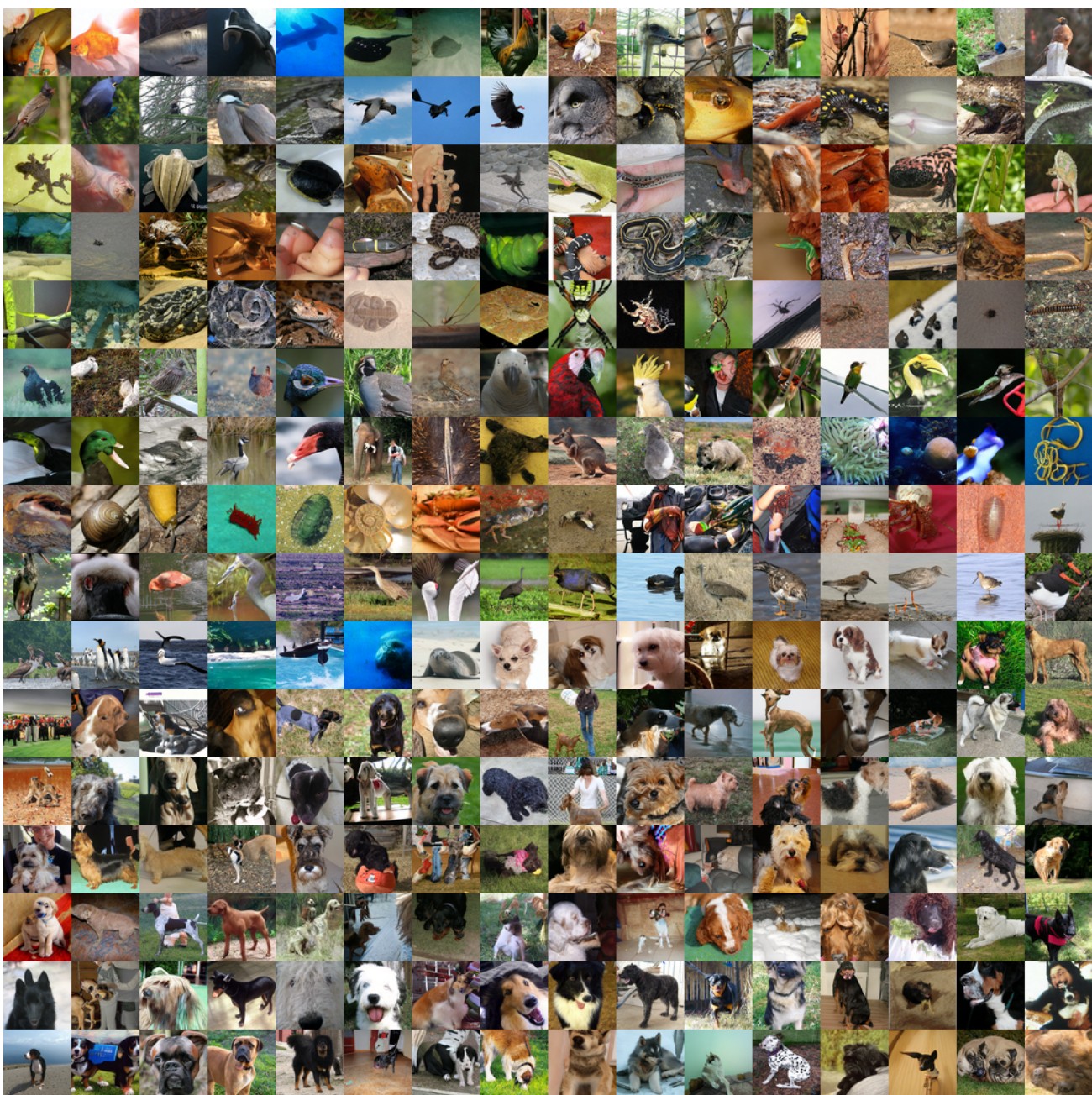

*Figure 11.* One-step generated samples from SMT on ImageNet 64×64 (conditional).

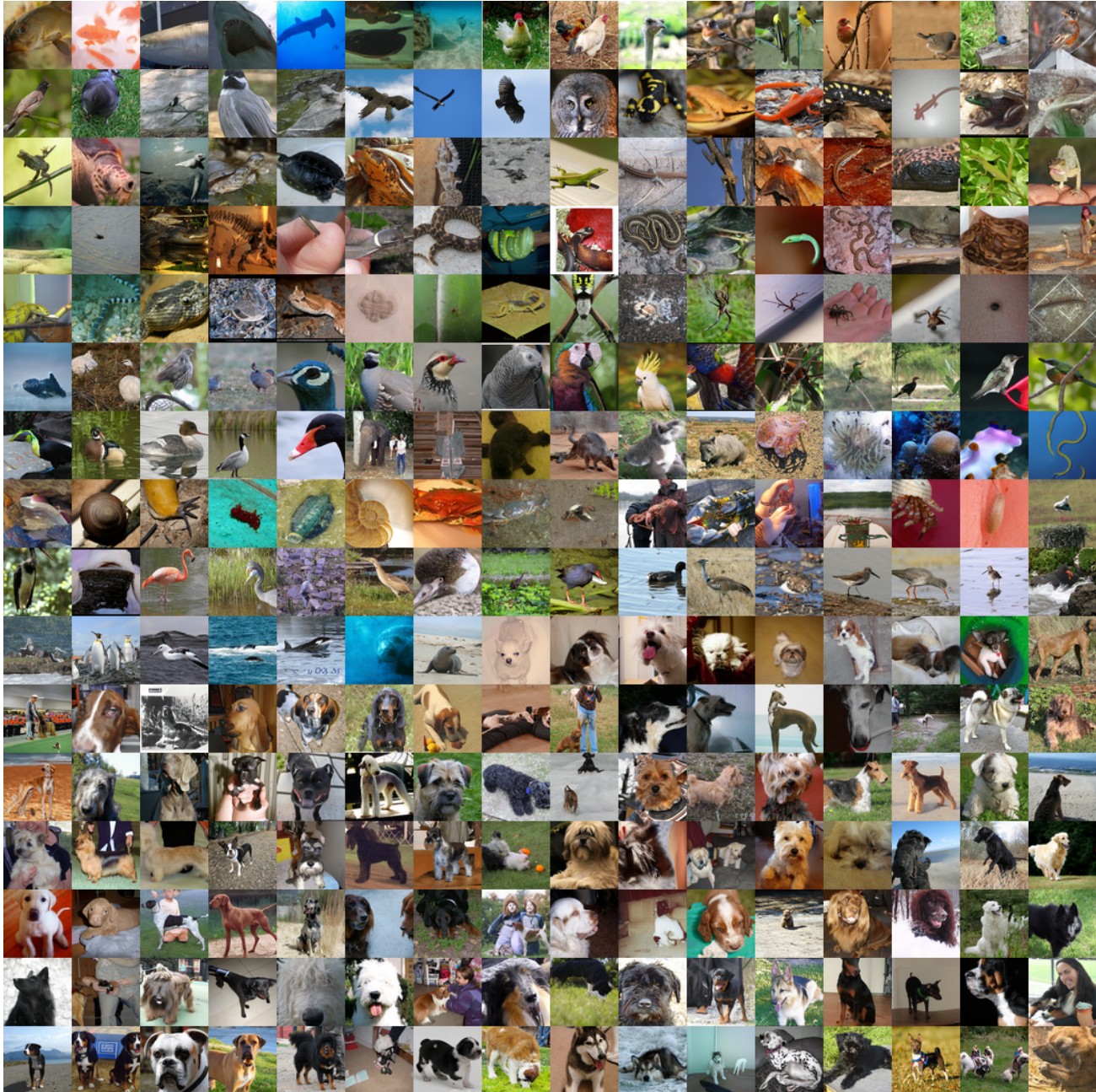

*Figure 12.* One-step generated samples from SMD on ImageNet 64×64 (conditional).

