# OpenReview forum: "Score-of-Mixture Training: One-Step Generative Model Training Made Simple via Score Estimation of Mixture Distributions"
_ICML.cc/2025/Conference — ICML 2025 spotlightposter_

### Official Review · Reviewer_cyoT · 2025-03-08

**Overall Recommendation:** 4

**Summary:**

This paper proposes a framework for training one-step generative models, called ScoreMix. The proposed method is derived by minimizing the $\alpha$-skew Jensen-Shannon Divergence ($\alpha$-JSD) between the generated distribution $q_{\theta}$ from an implicit generative model and the data distribution $p$ (or the generated distribution from the pretraining diffusion model in the distillation setting). The gradient of $\alpha$-JSD can be computed from the score of the mixture distribution of $q_{\theta}$ and $p$. Hence, training ScoreMix includes this mixture score network $s_{\psi}$ training. ScoreMix demonstrates competitive performance on  ImageNet 64x64 and CIFAR-10.

**Claims And Evidence:**

The claims regarding the methodology are generally well-supported, such as Prop 3.1, 3.2, and Cor 4.1

**Essential References Not Discussed:**

No

**Experimental Designs Or Analyses:**

I checked the validity of any experimental designs.

**Methods And Evaluation Criteria:**

One claimed advantage of ScoreMix in the manuscript is the stable training. However, this claim is not explicitly evaluated through empirical experimental results. Currently, it is supported only by assertions, such as the adoption of Multiple Noise Levels (Line 172) and denoising score matching techniques (Line 422)), without direct evidence. To substantiate this claim, the authors could consider various options, such as reporting the variance of scores or loss in Table 2 or Fig 2, or evaluating ScoreMix across diverse hyperparameters.

**Other Comments Or Suggestions:**

- Can we derive the relationship between the score of the mixture distribution $s_{\psi}$ between different values of $\alpha$ and utilize it as an additional regularizer?
- Could you clarify the 'expensive regularizers' in Line 189?

**Typo**
- Fig 1a in Line 381
- Fig 1b in Line 392.

**Other Strengths And Weaknesses:**

**Strength**
- This paper suggests a novel method for both training a one-step generator from scratch and distilling a pretrained diffusion model.

- The suggested approach is well-motivated by theoretical results.

- In Prop 3.2, estimating the mixture score using the mixture for the score matching loss is interesting.

- In Cor 4.1, the proposed method leads to a new way of training the discriminator in GAN.

**Weakness**
- Without the GAN regularizer, the performance of ScoreMIX is not competitive in Fig 2(b), which limits the novelty of the proposed method in Sec 3.1-3.4.

- The proposed method relies on GAN regularization in scratch training (Eq. 11), which might incur training instability as in GANs. Also, the distillation method requires three networks, i.e., generator, score network, and discriminator, which increases complexity.

- The proposed method requires initialization, such as warm-up training.

**Questions For Authors:**

Questions are included in other sections.

**Relation To Broader Scientific Literature:**

**Contributions**

- Novel training framework based on $\alpha$-JSD
- (Probably) stable training scheme motivated by DSM
- Supports both scratch training and distillation from a pretrained diffusion model
- Demonstrates competitive performance
- In Prop 3.2, estimating the mixture score using the mixture for the score matching loss is interesting.
- In Cor 4.1, the proposed method leads to a new way of training the discriminator in GAN.

**Theoretical Claims:**

I reviewed the proofs of Prop 3.1, 3.2, A.1, and A.2 presented in Appendix A.

---

> ### Author Rebuttal · Authors · 2025-04-01
>
> We appreciate the effort in reviewing our work and the helpful suggestions for improving the readability of our paper.
> Below, we provide clarifications on the identified weaknesses and responses to the questions.
>
> ### Clarifications on Weaknesses
> * `On stability of ScoreMix training`: We appreciate the reviewer’s suggestion to include training curves to further demonstrate the claimed stability of ScoreMix. While we included the FID curve over training iterations in Figure 2, we agree that plotting additional statistics would better illustrate training stability. In Figure D of this [anonymized link](https://docs.google.com/document/d/e/2PACX-1vTw0koYOvxXorKuyTBw3C4FPnB_tX6IE6sYZRUnuFhQfy2ixqVt5fDk5GjnDFlcInYUBy83hNKe10gT/pub), we have plotted an example training trajectory from our best ImageNet model, showing training losses and gradient norms. This further illustrates the stability during training. We will also include results for different parameter settings in the revision.
> * `On the role of the GAN regularizer`: We acknowledge that the GAN regularizer significantly helps accelerate convergence and improve FID. However, we note that we did not train the version without the GAN regularizer long enough to reach full convergence. To clarify whether the GAN regularizer is essential for achieving SOTA FID or primarily helps speed up convergence, we will conduct a training run on CIFAR10 without the GAN regularizer until convergence and report the result.
> * `On the stability and complexity of the GAN regularizer`: Similar to DMD2, our GAN discriminator is built on top of the score network, with only a few additional MLP layers. This score-model-dependent design allows the full model to benefit from the training stability provided by denoising score matching, while the GAN discriminator loss only trains the small auxiliary MLP. (For ImageNet, the generator has 296M parameters and the discriminator has 18M.) Thus, the discriminator represents a small fraction of the overall model size and has a negligible impact on training speed. As a result, our use of the GAN regularizer is both efficient and stable.
>
> ### Answers to Questions
> * `Additional regularization using consistency between scores of mixtures?`: We note that the score of a mixture distribution can be expressed as a weighted sum of the scores of the true and fake distributions, as shown in Eq. (12), where the weight is determined by the density ratio and $\alpha$. We leverage this relation in our distillation scenario, which is in the same spirit as the reviewer’s suggestion regarding consistency. However, for training from scratch, the relation between the scores for different $\alpha$ values is more implicit. It would be very interesting if a similar consistency regularization could be achieved in this context, and we agree that this is a promising direction for future work.
> * `On expensive regularizers`: We apologize for any confusion caused by our insufficient explanation of the term "expensive regularizers." We provide a clarification below and will address this point clearly in the revision.
>   * One example of an expensive regularizer is *the regression loss* used in the DMD paper. To address mode collapse in the DMD framework without any regularization, the authors simulate the reverse process of a diffusion model and sample several thousand noise-image pairs to anchor the generator’s outputs. Each noise-image pair requires evaluating the diffusion denoiser 256 times for ImageNet 64×64, which is extremely costly in practice. Moreover, the cost of collecting this regression dataset scales poorly with dimensionality.
>   * Another example of costly training is the approach in the CTM paper. To ensure consistency between random points along the PF ODE trajectory, the reverse diffusion sampler must be run for an arbitrary number of steps per minibatch, which also results in high computational cost. Here we note that we mistakenly referred to this expensive procedure as a "regularization" technique, and we will correct this terminology in the final revision.
>   * In contrast, our method does not require sampling from a diffusion model. It relies only on score estimation of mixture distributions, making it significantly more efficient.
>
> ---
>
> We thank the reviewer again for their insightful questions and helpful feedback. We will incorporate all of the above points in our revision, along with the proposed additional ablation studies.

---

> > ### Comment · Reviewer_cyoT · 2025-04-05
> >
> > Initially, I submitted an official comment that was not visible to the authors, so I am reposting it here:
> >
> > I appreciate the authors for the response and the additional Figure D, which supports the stable training dynamics of the proposed method. I am happy to raise my score from 3 to 4.

---

### Official Review · Reviewer_nETY · 2025-03-10

**Overall Recommendation:** 4

**Summary:**

This paper proposes a generalization of the KL-minimization procedure for learning one-step generators from score-based models. The authors introduce an "$\alpha$-skew Jensen–Shannon divergence", which interpolates between the KL divergence and the reversed-KL divergence. They propose two settings: one training from scratch, another one training with a pre-trained diffusion model. They add a GAN-based regularization strategy. Finally, they demonstrate with several experiments on image generation the interest of their method.

**Claims And Evidence:**

The paper mainly claims that minimizing the proposed $\alpha$-skew Jensen–Shannon divergence leads to better one-step generative models than minimizing the KL-divergence, which is a current practice in generative modelling. This is mainly evaluated in an ablation study on CIFAR-10 (Figure 1b), where the authors compare training with $\alpha=\{0,1\}$ and with random $\alpha$. I would also appreciate an ablation with standard KL or reversed KL divergence minimization in this Ablation study.

**Essential References Not Discussed:**

Not that I know.

**Experimental Designs Or Analyses:**

I checked the experimental design and analyses. As mentioned above, my biggest concern is with the use of the GAN regularization. Indeed, GAN regularization is not used in most other methods. It is thus a bit unfair to compare ScoreMix with other methods, since the other methods would also benefit from an adversarial loss. It could be interesting to include models trained without GAN regularization in Table 2. This would allow to assess the superiority of ScoreMix over other methods for one-step generator training.

Moreover, it would be interesting to compare the proposed GAN regularization with a standard GAN regularization to assess its interest.

Provided improvements on these crucial points, I would be willing to raise my score to Accept.

**Methods And Evaluation Criteria:**

Method: The method is sound, well-formulated, and elegant.

Evaluation criteria: The evaluation criteria is the FID on image generation benchmarks. The authors could consider relying on other type of metrics. Mainly, my biggest concern is the interaction between GAN losses and the FID. It has been shown that the FID is biased by adversarial losses (see "Exposing flaws of generative model evaluation metrics
and their unfair treatment of diffusion models", Stein et al., 2023). However, the ablation study shows that the improvement in FID does not only come from adversarial losses.

**Other Comments Or Suggestions:**

No

**Other Strengths And Weaknesses:**

No

**Questions For Authors:**

See weaknesses raised above (in Claim and Evidence, and in Evaluation Criteria).

**Relation To Broader Scientific Literature:**

The proposed paper is extending a widely used method for training one-step generators (KL-divergence minimization with score-based models). It proposes the minimization of $\alpha$-skew JSD, which interpolates between KL divergence and reversed KL divergence. This is new and original to me.

**Theoretical Claims:**

The theoretical claims mostly extend known results to $\alpha$-skew JSD. I did not check the proofs in detail.

---

> ### Author Rebuttal · Authors · 2025-04-01
>
> We appreciate the reviewer’s effort in reviewing our paper and constructive comments. We will address the raised concerns in our revision as follows.
>
> * `On FID evaluation and adversarial loss`: We appreciate the reviewer for the thoughtful comment on the limitation of FID evaluation and its interaction with adversarial losses (Stein et al., 2023). We will clearly explain these points in our revision. Here, we wish to clarify our standpoint.
>   * While we acknowledge that FID is not a perfect metric as argued in the reference, it remains the most popular benchmark for measuring perceptual quality. Our ablation study shows that ScoreMix achieves stable and competitive performance even without the GAN regularization, indicating that the performance improvement is not solely due to adversarial losses.
>   * Regarding fairness in comparison, we highlight that some recent distillation methods in Table 2 such as DMD2 use adversarial regularizer. Similar to DMD2, our GAN discriminator is built on top of the score network, with a few layers of MLP. This score-model-dependent design allows the entire model to enjoy training stability driven by denoising score matching, while the GAN discriminator loss only trains the small additional MLP. (For ImageNet, the generator has 296M params and the discriminator has 18M params.) However, for other distillation or training-from-scratch methods such as consistency training or distillation, it is nontrivial to implement the GAN discriminator assuring training stability. For example, the notable example of the consistency model framework differs conceptually from traditional distribution matching approaches. While a GAN regularizer fits within this framework, its implementation requires more than a small discriminator like ours. Designing a full discriminator network (similar to those in StyleGAN or its XL variants) would be necessary, and this could involve substantial hyperparameter tuning and careful initialization. Moreover, due to the inherent instability of the consistency training framework, introducing an adversarial loss could exacerbate these instabilities, given the well-known challenges of GAN training. Hence, investigating GAN regularization to an existing method such as consistency training is an interesting research direction, but it is beyond our current scope.
> * `Comparison of the proposed GAN regularization with standard GAN regularization`: This will be an informative ablation study that can clarify the role of skew divergence in GAN regularizer. We will add an additional result for CIFAR10 with standard GAN regularization, i.e., only using $\\alpha=\\frac{1}{2}$ in the regularization.
> * `Additional ablation with standard KL or reversed KL divergence minimization`: We appreciate the reviewer’s suggestion. We will add a result only using the reverse KL divergence for CIFAR10 in the revision. In the meantime, we performed an additional experiment with a toy dataset, as shared in the global response.
> * `Toy experiment on Swiss roll dataset`: We ran a toy experiment on 2D Swiss roll dataset, to demonstrate the effectiveness of our ScoreMix framework compared to the existing schemes for a simpler setting; see Figure A in this [anonymized link](https://docs.google.com/document/d/e/2PACX-1vTw0koYOvxXorKuyTBw3C4FPnB_tX6IE6sYZRUnuFhQfy2ixqVt5fDk5GjnDFlcInYUBy83hNKe10gT/pub). In particular, our results for training from scratch and distillation are presented in Figure A (d, f, g). All three methods successfully capture the modes of the underlying distribution. While the impact of the GAN regularizer is less pronounced than in our high-dimensional experiments, we observe that enabling it reduces the number of samples in low-density regions in (g). The distillation results in (d) appear slightly noisy, likely due to the quality of the pre-trained score model. This highlights the advantage of training from scratch, as it avoids amplifying existing estimation errors in the pre-trained model. We will add this experiment to Appendix in our revision.
>
> We thank the reviewer again for their suggestions and comments, and will certainly incorporate all the points in our revision including additional ablation study results with CIFAR10. If these clarifications satisfactorily address the reviewer's concerns, we kindly ask if the reviewer would consider updating the score to reflect what we believe is a paper with noteworthy contributions to the community.

---

### Official Review · Reviewer_vjzy · 2025-03-14

**Overall Recommendation:** 4

**Summary:**

The paper presents the ScoreMix, a new type of one-step generative model, trained using the $\alpha$-JSD from $f$-divergence. ScoreMix can be trained from scratch and used for distillation. It achieves SOTA performance in the 1-NFE regime. The paper grounds the theoretical approach and performs extensive experiments/analyses to showcase its significance.

**Claims And Evidence:**

All the claims made in this paper are supported by clear and convincing evidence.

**Essential References Not Discussed:**

Rectified flow models and their variations also fall under the 1NFE requirement. Hence, for a better comprehensive understanding, they should be included in Table 2.

[1] "Flow Straight and Fast: Learning to Generate and Transfer Data with Rectified Flow"
[2] "Improving the Training of Rectified Flows"
[3] "One Step Diffusion via Shortcut Models"

**Experimental Designs Or Analyses:**

Yes, the soundness of the following experimental designs by the author is verified and looks good.

**Methods And Evaluation Criteria:**

Image generation is a testbed for evaluating generative models (especially for diffusion/flow models). The authors follow standard evaluation criteria that evaluate the method on the community-defined datasets (ImageNet64 & CIFAR10).

**Other Comments Or Suggestions:**

- Instead of Fig. 2b, many sentences (Lines 256, 387, 392, 409) contain the typo and mention Fig. 1a.

**Other Strengths And Weaknesses:**

Strengths:
- The paper has been written concisely and clearly.
- ScoreMix achieves the SOTA performance on the ImageNet64 and CIFAR10 that can be trained from scratch and contains the ablation on $\alpha$ parameter that clearly showcases the advantage of the design choices made for ScoreMix.

Weaknesses:

Overall, the paper is in good shape, except for some minor improvements that could be made.
- **Missing baselines:** Rectified Flow models and their variants, such as RF++ and ShortCut models, could be included as part of the main results in Table 2.
- **Efficiency comparisons:** Although it is claimed that the training budget is smaller for ScoreMix, it is important to share the total GPU hours for each experiment and how much it does for at least some baselines. Otherwise, training the two models seems a slow and expensive process.
- **Code release**: Authors claim to release the code to the public after the reviewing process; however, they have not even shared it as a part of supplementary material for reviewers. Hence, it is hard to verify the actual reproducibility. That said, I trust the authors will release it; hence, my current rating is also conditionally.

**Questions For Authors:**

- Could authors elaborate more on why $\alpha=0$ for 25% of the time during the score training (Lines 261-263)?
- ScoreMix currently supports only the 1NFE approach. Can authors share some insights (if any) on how this could be extended to multi-step inference, as many downstream tasks such as editing and inverse problems might depend on it?
- Is it possible to perform inversion at all?
- What does the linear interpolation between two random input noises result in? Take two noises, then get interpolated noises in-between, and create a gif or image collage of corresponding generations. This could shed some light on the latent space of ScoreMix.

**Relation To Broader Scientific Literature:**

This paper presents the new type of generative model that is 1NFE and seems easy to train and could be of interest to various communities such as audio, applied, and AI4Science.

**Theoretical Claims:**

The reviewer has not carefully verified derivations line-by-line from the appendix. However, the reviewer believes the idea is straightforward and intuitively makes sense.

---

> ### Author Rebuttal · Authors · 2025-04-01
>
> We appreciate the effort in reviewing our manuscript and providing constructive comments. We will incorporate all the feedback in our revision to improve the manuscript.
>
> ### On Weaknesses
>
> * `Missing reference and baselines`: We thank the reviewer for pointing out the missing references and baselines (rectified flow models and their variants). We will certainly add and discuss these in our revision to better clarify and contextualize our contribution.
> * `Efficiency comparisons`: We thank for the suggestion. We will include GPU hours for our experiments in the revision to highlight the efficiency of our framework. To obtain the best FID result for ImageNet reported in Table 2, for example, it took 80 hours using 7 × A100 GPUs (200k iterations with an overall batch size of 280). As a comparison, both consistency training (CT) and improved consistency training (iCT) used 800k iterations with a much larger batch size of 2048. They do not provide the number of GPU hours, making a direct comparison difficult. We do note that ECT is able to *finetune* a pre-trained diffusion model on 4 × H100 GPUs within 8.5 hours. However, this consistency model is initialized with a pre-trained EDM2 diffusion model that was trained on 32 × A100 GPUs with a batch size of 2048 for a total of 2500 million images. Again, the EDM2 paper does not report the number of GPU hours, but its predecessor, EDM, reports a total training time of 2 weeks with a similar computational budget on ImageNet 64×64 — which alone exceeds our entire training-from-scratch effort. This shows that our proposed method can train a one-step model from scratch with a far smaller computational budget in terms of number of GPUs, training iterations, and overall batch size. Moreover, the two-model update is not more expensive than existing SOTA methods.
> * `Code release`: As promised, we will release our codebase and model weights upon acceptance for reproducibility.
> * `Typo`: We appreciate the reviewer’s careful reading of our paper. We will correct the typos and thoroughly check the manuscript.
>
> ### Answers to Questions
>
> * `Why α=0 for 25% of the time during the score training?`: This is due to the nature of the gradient of the skew JSD in Eq. (4). In that expression, the score of the mixture distribution with $\alpha=0$ is *always* used, which implies its particular importance and the need for accurate estimation. We will clarify this point in our revision to avoid any confusion.
> * `Extension to multi-step inference?`: This is a great question, and we agree that having a multi-step refinement feature would be of great interest. One potential direction is to develop a hybrid method that combines consistency models with our approach. We leave this for future work.
> * `Invertibility?`: Our current manuscript focuses on developing a high-quality, efficient sampling scheme. Whether our model can be inverted is indeed an interesting question from a representation learning perspective, and we leave this for future investigation.
> * `Linear interpolation?`: Thanks for the helpful suggestion. We conducted the interpolation experiment for CIFAR10 and ImageNet and uploaded the results at this [anonymized link](https://docs.google.com/document/d/e/2PACX-1vTw0koYOvxXorKuyTBw3C4FPnB_tX6IE6sYZRUnuFhQfy2ixqVt5fDk5GjnDFlcInYUBy83hNKe10gT/pub); see Figures B and C. Similar to GANs and consistency models, we found that “spherical interpolation” leads to natural interpolations.

---

> > ### Comment · Reviewer_vjzy · 2025-04-01
> >
> > I thank the authors for providing a detailed response and clarifications.
> >
> > Diffusion/flow/consistency models have been explored to solve inverse problems (inpainting, deblurring, super-resolution, etc.) [1-3]. Do authors believe that ScoreMix can help in such problems, especially because 1NFE inference could improve the convergence rate?
> >
> > **Overall, I like the current draft and rebuttal. Happy to maintain my current score.**
> >
> > [1] Ben-Hamu et al., "D-Flow: Differentiating through Flows for Controlled Generation"
> > [2] Chung et al., "Diffusion Posterior Sampling for General Noisy Inverse Problems"
> > [3] Patel et. al., "Steering Rectified Flow Models in the Vector Field for Controlled Image Generation"

---

> > > ### Author Response · Authors · 2025-04-07
> > >
> > > Thank you once again to the reviewer for their insightful feedback and comments. We apologize for the delayed response to your most recent question.
> > >
> > > We agree that the ScoreMix framework holds promise for solving inverse problems, and this is a key focus of our ongoing and future research. In works like DPS [2], the primary quantity of interest is the posterior expectation $p(\mathbf{x} | \mathbf{y})$, or more formally, its score. While DPS approximates this quantity by leveraging a pre-trained denoiser, recent studies [4, 5, 6] have explored using distribution matching frameworks based on distillation to train a posterior score model, along with a generator capable of sampling from this posterior distribution in just one NFE. We believe that our ScoreMix-distillation framework can be similarly adapted to address this problem.
> > >
> > > To the best of our knowledge, there are very few, if any, frameworks that allow for training a 1NFE posterior sampler from scratch. A naive extension of our training-from-scratch approach to minimize the skewed Jensen-Shannon divergence between the true posterior $p(\mathbf{x} | \mathbf{y})$ and the fake posterior $q_\theta(\mathbf{x} | \mathbf{y})$ results in a tractable gradient for updating the generator. However, the amortized score estimation loss would need to be augmented to allow for computing expectations over $p(\mathbf{x} | \mathbf{y})$. This represents an intriguing extension, and we believe that developing a solution to it could provide new insights into training the amortized score model.
> > >
> > > [4] Mammadov, Abbas, Hyungjin Chung, and Jong Chul Ye. "Amortized Posterior Sampling with Diffusion Prior Distillation." arXiv preprint arXiv:2407.17907 (2024).
> > >
> > > [5] Wu, Zihui, et al. "Principled probabilistic imaging using diffusion models as plug-and-play priors." Advances in Neural Information Processing Systems 37 (2024): 118389-118427.
> > >
> > > [6] Lee, Sojin, et al. "Diffusion prior-based amortized variational inference for noisy inverse problems." European Conference on Computer Vision. Cham: Springer Nature Switzerland, 2024.

---

### Decision · Program_Chairs · 2025-05-01

**Decision:**

Accept (spotlight poster)

**Comment:**

This paper proposes a new method for training one-step generative models based on the $\alpha$-skew Jensen-Shannon divergence. Reviewers unanimously praised the writing, novelty, and empirical results of the paper. The method is sound and the empirical results are strong, and I believe this paper might be impactful in the area. I thus recommend acceptance.